# Designing transparent piezoelectric metasurfaces for adaptive optics

Liao Qiao[1], Xiangyu Gao [1] ✉, Kaile Ren [1], Chaorui Qiu[1], Jinfeng Liu[1], Haonan Jin[1], Shuxiang Dong[2], Zhuo Xu[1] ✉ & Fei Li [1] ✉

Simultaneously generating various motion modes with high strains in piezo-electric devices is highly desired for high-technology fields to achieve multi-functionalities. However, traditional approach for designing multi-degrees-of-freedom systems is to bond together several multilayer piezoelectric stacks, which generally leads to cumbersome and complicated structures. Here, we proposed a transparent piezo metasurface to achieve various types of strains in a wide frequency range. As an example, we designed a ten-unit piezo metasurface, which can produce high strains ($\varepsilon_3 = 0.76\%$), and generate linear motions along X-, Y- and Z-axis, rotary motions around X-, Y- and Z-axis as well as coupled modes. An adaptive lens based on the proposed piezo metasurface was demonstrated. It can realize a wide range of focal length (35.82 cm ~ ∞) and effective image stabilization with relatively large displacements (5.05 μm along Y-axis) and tilt angles (44.02′ around Y-axis). This research may benefit the miniaturization and integration of multi-degrees-of-freedom systems.

Piezoelectric devices have been widely employed in robot vision[1,2], precise optical instruments[3–5], health diagnosis[6,7], consumer electronics[8,9], and micro electromechanical systems[10,11]. Compared with traditional electromagnetic actuators, piezoelectric devices exhibit the merits of compact size, no electromagnetic interference, low noise and fast dynamic response, thus have received widespread attention in recent decades[12–15]. Furthermore, under the external electric field, piezoelectric materials can generate stresses and strains through inverse piezoelectric effect, which can produce a variety of vibration modes.

To implement multi-functional interaction modes in a piezo-electric device, substantial efforts have been made[13,14,16–20]. The classic approach is to bond together several multilayer piezoelectric stacks that originally only produces the axial displacement in a certain direction to construct multi-degrees-of-freedom systems[16–18]. For example, a 3-degrees-of-freedom manipulator requires at least three multilayer piezoelectric stacks, including one with longitudinal mode (33-mode) and the other two with orthogonal shear modes (15-mode)[16]. However, the structure of such multi-degrees-of-freedom system is generally very cumbersome and complicated. To achieve integration and miniaturization, Li et al. [13] proposed a 3D ordered

structure with piezoelectric ceramic units based on synergic strain effect, which realized artificial vibration modes (including stretching, shear, bending and torsion modes) in the piezoelectric ceramics. The 3D ordered structure can produce actuations with 5-degrees-of-freedom under program-controlled voltages. However, lots of elec-tromechanical devices, such as piezoelectric motors, require coupled vibration modes to provide an interactive mode with multifunction. In the above-mentioned research, various coupled vibration modes cannot simultaneously achieve, since different piezoelectric vibration modes had to be induced under different boundary conditions. For example, as mentioned in ref. 13, the artificial torsion mode around 1-axis and 2-axis were achieved under fixed constrains in the planes normal to 1- and 2-axis, respectively. To solve this issue, Liu et al.[14] designed a metamaterial with all non-zero piezoelectric coefficients based on the uniform strain units under stress free boundary condition of a topological structure. However, different unit arrangements and different electric field directions were adopted to produce the desired vibration modes. Furthermore, to induce the desired vibration modes and suppress the parasitic modes, the electric field with pre-determined frequency (i.e., characteristic resonance frequencies) had

[1]Electronic Materials Research Laboratory, Key Lab of Education Ministry and State Key Laboratory for Mechanical Behavior of Materials, School of Electronic Science and Engineering, Xi'an Jiaotong University, Xi'an 710049, China. [2]Institute for Advanced Study, Shenzhen University, Shenzhen 518051, China. ✉e-mail: gaoxiangyu@xjtu.edu.cn; xuzhuo@xjtu.edu.cn; ful5@xjtu.edu.cn

to be applied. Therefore, to satisfy the practical needs of piezoelectric devices, simultaneously accomplishing various vibration modes from piezoelectric units under a uniform boundary condition and a wide frequency range is highly required.

Herein, a transparent piezo metasurface (PM) is proposed to simultaneously achieve desired motion modes with high strains in a wide range of frequencies, including the linear motions along the X-, Y- and Z-axis, the rotary motions around the X-, Y- and Z-axis and the coupled motions. Through the design of topological structures, unit dimensions and boundary structures for PM, we demonstrate that the various types of high strains ($\varepsilon_3 = 0.76\%$) can be obtained in a wide frequency range via multiple various motion modes. To practically realize this idea, we employ transparent [001]-poled Pb(In$_{1/2}$Nb$_{1/2}$)O$_3$-Pb(Mg$_{1/3}$Nb$_{2/3}$)O$_3$-PbTiO$_3$ (PIMNT) single crystals as piezoelectric units, and design an adaptive lens (ALENS) consisting of ten pieces of piezoelectric units. By using only one PM, the ALENS can simultaneously achieve both adjusting focus (AF) and optical image stabilization (OIS) functions. Our design strategy of the high-performance PM may promote the design of multi-functions intelligent devices in microelectromechanical systems.

## Results and discussion

### Design of PM

Piezo metasurface is a two-dimensional piezoelectric material consisting of periodic piezoelectric units, which generates multiple motion modes by applying electric fields. To clarify the design for PM, Fig. 1a displays the basic elements of piezo metasurfaces, including the basic piezoelectric unit and deformations, where the piezoelectric materials are selected as units and arranged in the X-Y plane via our designed center outward expansion method to construct the PM. The center outward expansion method is an orderly structure construction method in which piezoelectric units expand outward circle by circle from the center of the structure with a consideration of required motion modes. Specifically, $A_{10}$ ($A_{20}$) unit is in the center, $A_{11}$, $A_{12}$, $A_{13}$, and $A_{14}$ ($A_{21}$, $A_{22}$, $A_{23}$ and $A_{24}$) units are the first circle, and $A_{15}$, $A_{16}$, $A_{17}$ and $A_{18}$ ($A_{25}$, $A_{26}$, $A_{27}$ and $A_{28}$) units are the second circle, and so on (Fig. 1 and Supplementary Fig. 1). With applying the AC voltages, the specific region of the PM can generate transverse-extension deformation or axial-bending deformation (Fig. 1a). Based on the synergistic strain effect among the piezoelectric units[13,14], the PM can generate greatly enhanced deformation along a specific direction to achieve the desired motion modes.

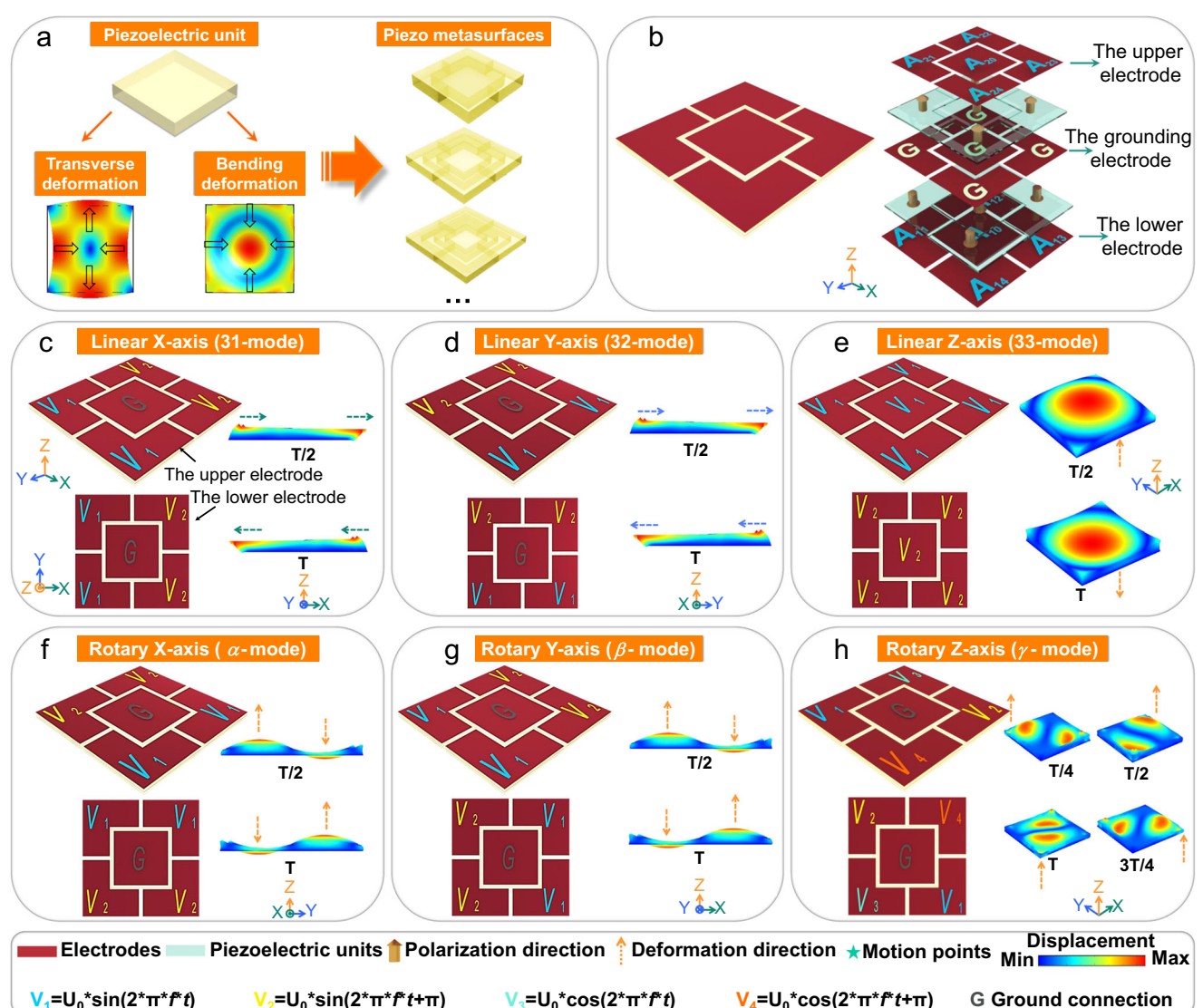

**Fig. 1 | Design of a PM to generate the desired motion modes. a** Designed elements of a PM. **b** Exploded figure of an (5 × 2) arrayed PM. The applied voltages and simulated deformations of (**c**) the linear motion along the X-axis (artificial 31-mode), (**d**) the linear motion along the Y-axis (artificial 32-mode), (**e**) the linear motion along the Z-axis (artificial 33-mode), (**f**) the rotary motion around the X-axis ($\alpha$-mode), (**g**) the rotary motion around the Y-axis ($\beta$-mode), (**h**) the rotary motion around the Z-axis ($\gamma$-mode).

To verify the feasibility of the methodology, an (5 × 2) arrayed PM is designed to generate the desired motion modes in a wide range of frequencies under one or multiple voltages, including the linear motions along the X-, Y- and Z-axis, the rotary motions around the X-, Y- and Z-axis and the coupled motions. As shown in Fig. 1b, the PM is composed of ten piezoelectric units, and each unit is distinguished by the subscript numbers, where the first subscript number 1 or 2 is denoted as the first-layer or second layer, respectively. For the second subscript number, the central unit is denoted by 0, and along clockwise around the center unit in order of 1, 2, 3 and 4. The polarization direction of the five units ($A_{10}$, $A_{11}$, $A_{12}$, $A_{13}$ and $A_{14}$) on the first-layer is along the negative Z-axis, while the polarization direction of the five units ($A_{20}$, $A_{21}$, $A_{22}$, $A_{23}$ and $A_{24}$) on the second-layer is along the positive Z-axis. In this design, all units share a common ground electrode. As the applied electric fields are parallel or antiparallel to the polarization direction of each unit in the PM, the linear motions or the rotary motions can be generated. The linear motions correspond to transverse-extension vibration modes ($\varepsilon_j$, j = 1, 2) and the longitudinal-extension vibration mode ($\varepsilon_j$, j = 3), and the rotary motions correspond to the rotational degrees of freedom ($\alpha$, $\beta$ and $\gamma$).

With the guidance of the piezo metasurfaces methodology, we can acquire the linear motions and the rotary motions of the PM. To induce a transverse-extension deformation in the PM, piezoelectric units with 31- or 32-mode are used to construct the transverse-extension topological structure (Fig. 1c, d). As shown in Fig. 1c, for the linear motion along the X-axis, the $A_{11}$, $A_{14}$, $A_{21}$ and $A_{24}$ units with voltage signal $V_1$ are classified as group I, while the $A_{12}$, $A_{13}$, $A_{22}$ and $A_{23}$ units with voltage signal $V_2$ are classified as group II, where the phase difference of AC voltage signals $V_1$ and $V_2$ is π. Then, opposite transverse-extension deformations can be produced for group I and II, i.e., group I elongates (or contracts), while group II contracts (or elongates). The whole PM displays a lateral translation deformation along the X-axis, corresponding to the linear motion along the X-axis (artificial 31-mode). Similar situation is also present in Fig. 1d, corresponding to the linear motion along the Y-axis (artificial 32-mode).

To induce an axial-bending deformation in the PM, piezoelectric units with 31- or 32-mode are used to construct the longitudinal-bending topological structure (Fig. 1e–h). As shown in Fig. 1e, for the linear motion along the Z-axis, the $A_{10}$, $A_{11}$, $A_{12}$, $A_{13}$ and $A_{14}$ units with voltage signal $V_1$ are classified as group I, while the $A_{20}$, $A_{21}$, $A_{22}$, $A_{23}$ and $A_{24}$ units with the voltage signal $V_2$ are classified as group II, where the phase difference of AC voltage signals $V_1$ and $V_2$ is π. In this case, group I and II can be excited with opposite deformation: group I produces an expansion (or contraction) deformation, while group II produces a contraction (or expansion) deformation, leading to the arching (or incurvaing) deformation along the positive (or negative) direction of the Z-axis. As a result, the whole PM displays an integral bending deformation in the X-Y plane (parallel to the Z-axis), corresponding to the linear motion along the Z-axis (artificial 33-mode). In addition, when only the central units ($A_{10}$ and $A_{20}$) were used to induce arching (or incurvaing) deformation along the Z-axis, this mode is noted as artificial 33*-mode.

As shown in Fig. 1f, for the rotary motion around the X-axis ($\alpha$-mode), the $A_{11}$, $A_{12}$, $A_{23}$ and $A_{24}$ units with the voltage signal $V_1$ are classified as group I, while the $A_{13}$, $A_{14}$, $A_{21}$ and $A_{22}$ units with the voltage signal $V_2$ are classified as group II, where the phase difference of AC voltage signals $V_1$ and $V_2$ is π. In detail, the axial-bending deformation could be produced, i.e., the left half of the units ($A_{11}$, $A_{12}$, $A_{21}$ and $A_{22}$ units) induces arching (or incurvaing) deformation along the positive (or negative) Z-axis, while the right half of the units ($A_{13}$, $A_{14}$, $A_{23}$ and $A_{24}$ units) induces incurvaing (or arching) deformation along the negative (or positive) Z-axis. Thus, the whole PM displays the rotational motion mode around the X-axis in the X-Y plane, corresponding to the rotary motion around the X-axis ($\alpha$-mode). Similar situation is also present in Fig. 1g, corresponding to the rotary motion around the Y-axis ($\beta$-mode).

As shown in Fig. 1h, for the rotary motion around the Z-axis ($\gamma$-mode), the $A_{21}$ and $A_{13}$ units with voltage signal $V_1$ are classified as group I, the $A_{22}$ and $A_{14}$ units with voltage signal $V_2$ are classified as group II, the $A_{23}$ and $A_{11}$ units with voltage signal $V_3$ are classified as group III, the $A_{24}$ and $A_{12}$ units with voltage signal $V_4$ are classified as group IV, where the phase difference of AC voltage signals $V_1$, $V_2$, $V_3$, $V_4$ is π/2. Then, the axial-bending deformation could be produced for group I, II, III and IV along the positive (or negative) Z-axis in turn under AC voltage. The whole PM eventually observes torsional motion around the Z-axis in the X-Y plane, corresponding to the rotary motion around the Z-axis ($\gamma$-mode). Supplementary Movie 1 presents that the dynamic deformation graphics of all the desired motion modes for the PM obtained by FEM simulation. More coupled motion modes can be realized by changing the arrangement, size, or number of piezoelectric units. For example, according to the center outward expansion method, the structure 5 × 2, 9 × 2 and 21 × 2 arrayed PMs are shown in Supplementary Figs. 2–4. In addition, test positions of the desired motion modes are presented in Supplementary Fig. 5.

## The simulation results of PM

The output performance of the desired motion modes for the PM involves three factors: intrinsic properties of piezoelectric materials, boundary structures and geometric dimensions. The linear motions and rotary motions of the PM correspond to the piezoelectric strains ($\varepsilon_1$, $\varepsilon_2$ and $\varepsilon_3$) and the rotation angles ($\alpha$, $\beta$, and $\gamma$), respectively. The $\varepsilon_1$, $\varepsilon_2$, $\varepsilon_3$, $\alpha$, $\beta$, and $\gamma$ values can be calculated according to Supplementary Note 1. In the following, we will optimize the output performance of the PM in terms of the above three aspects, as summarized in Fig. 2. The insets are the simulated deformations of the linear motions and rotary motions of PM. The [001]-poled PIMNT single crystals[21] and soft PZT ceramics[22] are selected as typical materials.

Firstly, the output performance (the piezoelectric strain $\varepsilon_3$ of the artificial 33-mode) for the PM is optimized in terms of the Young's modulus, structures and geometry dimensions of the boundary (Fig. 2a–d). (i) The output strains $\varepsilon_3$ of the artificial 33-mode with respect to the Young's modulus of the boundary are represented in Fig. 2a. As expected, the $\varepsilon_3$ values increased as the Young's modulus decreased. In our design, the Young's modulus of the boundary is set to be 750 kPa. It is worth noting that if the Young's modulus of the boundary structure is too low, it is prone to deformation and thus cannot support the crystals to produce desired motion modes. (ii) As shown in Fig. 2b, there are significant differences in $\varepsilon_3$ values among the different boundary structures. The boundary structures can be found in Fig. 2b and Supplementary Fig. 7, where the boundary structures of the PM are distinguished in the following by the number of layers of the boundary, including $PM_0$, $PM_1$, $PM_2$ and $PM_3$. The structure of boundary optimized is more reasonable, with its output performance ($\varepsilon_3$ values) increased greatly. Compared to the other boundary structures, the $PM_2$ possess the highest output strain (0.405%) because the stress distribution of the $PM_2$ tends to be uniform and stays at a relatively high level after optimization (Supplementary Table 1). (iii) The size optimization of the single crystals $PM_2$ is adopted, including $H_{HS}$, $L_{HS}$ and $W_{HS}$. The $\varepsilon_3$ value increased with increasing $H_{HS}$ and $L_{HS}$, while it decreased with the increase in $W_{HS}$, as given in Fig. 2c. Then, we set $H_{HS}$, $L_{HS}$ and $W_{HS}$ to be 4.8 mm, 13 mm and 0.5 mm, respectively. Here, the picture of the boundary structure is given in Fig. 2d, which is fabricated by 3D printing (DPI8400, Dongguan Mold Material Co., Ltd.).

Secondly, the output performance for the PM as a function of the length to thickness ratio (LTR) under the electric field of 400 V/mm was given in Fig. 2e–h. Since the performance variation of the artificial 31-mode and 32-mode, $\alpha$-mode and $\beta$-mode with LTR are the same, the size optimization of the artificial 32-mode and $\beta$-mode are not described here to avoid repetition. The results show that the effective values ($\varepsilon_1$, $\alpha$, and $\gamma$) increases linearly, as the LTR increases, while $\varepsilon_3$

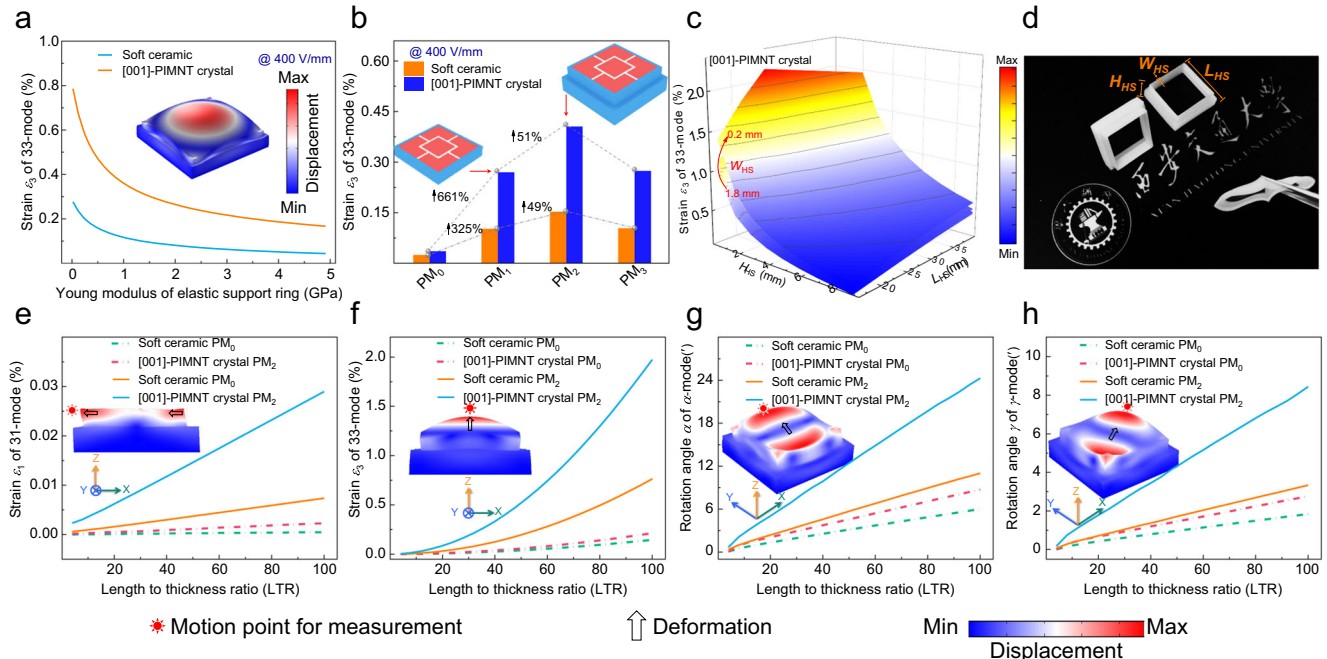

**Fig. 2 | Output performance for the desired motion modes as a function of the boundary structures and geometric dimensions of the PM by FEM simulation.** **a** The strain $\varepsilon_3$ variation of the PM as a function of the Young's modulus of the boundary structures. **b** The strain $\varepsilon_3$ variation of artificial 33-mode with the boundary structures. (The insets show the boundary structures of $PM_1$ and $PM_2$.) **c** The strain $\varepsilon_3$ variation of artificial 33-mode with the geometric dimensions of the boundary structure. **d** Photograph of the boundary structure. The strain ($\varepsilon_j$, j = 1 and

3) variation of (**e**) artificial 31-mode and (**f**) artificial 33-mode with the geometric dimensions of the piezoelectric units for the PM. The rotation angle ($\alpha$ and $\gamma$) variation of (**g**) $\alpha$-mode and (**h**) $\gamma$-mode with the geometric dimensions of the piezoelectric units for the PM. The insets show the geometrical diagrams and deformations of the simulated motion modes by using $PM_2$. The black arrows represent displacement directions. The electric field of 400 V/mm was applied on the soft PZT ceramics and [001]-poled PIMNT single crystals.

seems to increase nonlinearly. The effective values can be controlled over a broad range by resizing the LTR of the PM. Meanwhile, the effective values for the $PM_2$ are one order larger than that of the $PM_0$ under the same condition, and their values can be further improved using the [001]-poled PIMNT single crystals compared to the soft PZT ceramics. For example, the $\varepsilon_3$ for the [001]-poled PIMNT crystals $PM_2$, the soft PZT ceramics $PM_2$, the [001]-poled PIMNT crystals $PM_0$, and the soft PZT ceramics $PM_0$, as LTR = 100, are, respectively, 1.97%, 0.76%, 0.21%, and 0.15%. It is noted that $PM_2$ is further discussed in the following text and named PM.

**The experimental results of PM**
We fabricated two piezo metasurfaces by using transparent [001]-poled PIMNT single crystals and soft PZT ceramics. The vibration displacements were measured under different electric fields at 2 Hz, as shown in Fig. 3. The experimental setup demonstrated in Fig. 3a. The dimensions of the PM with the ceramics and crystals are both designed as 10 mm$^{Length}$ × 10 mm$^{Width}$ × 0.25 mm$^{Thickness}$ (LTR = 40). The output displacements remarkably correlated linearly with the operating electric fields at 2 Hz in the range of 0 - 800 V/mm, which are in good accordance with the simulation data. Significantly, for the PM made of PIMNT crystals, the experimental displacement values of the artificial 31- and 32-mode are just slightly below the simulated results, while the measured values of the artificial 33-mode are noticeably greater than the simulation, while the test results of the $\alpha$-, $\beta$-, and $\gamma$-mode are similar to the simulated values. These differences are attributed to imperfect boundary structures and inhomogeneous deformation at the surface of the PM. In addition, the piezoelectric strain coefficients ($\varepsilon_1$, $\varepsilon_2$ and $\varepsilon_3$) and the rotation angles ($\alpha$, $\beta$, and $\gamma$) of the motion modes can be calculated with Supplementary Eqs. (S1)−(S6) in Supplementary Note 1, and their corresponding values, including the simulation and experiment values under the electric field of 800 V/mm, are listed in Table 1 and Supplementary Table 2. The experimental data $\varepsilon_1$, $\varepsilon_2$ and $\varepsilon_3$

of the PM made of crystals are calculated to be 0.022%, 0.023% and 0.756%, respectively, which are close to the simulation values of 0.029%, 0.029%, and 0.666%. Significantly, the strain $\varepsilon_3$ of the artificial 33-mode (0.756%) is about more than one order of magnitude higher than the natural value of crystals (0.054%)[21].

**Design and characterization of the PM-based ALENS**
To show the advantages of our proposed PM in practical applications, we designed a PM-based ALENS with AF and OIS functions. Thanks to the linear motions, the rotary motions and the coupled motions with high strains, the PM is a promising candidate for the adaptive optics systems that require compact structure and functional integration. Traditional ALENS generally requires multiple actuators to achieve AF and OIS functions[23,24]. Here, the AF and OIS functions can be achieved using only one PM. Figure 4 depicts the schematic and structure of the PM-based ALENS, where the PM is composed of ten pieces of transparent PIMNT single crystal units with ultrahigh piezoelectricity. In addition, Euler angles are adopted to describe the rotary motions, including pitch (the rotary motion around the X-axis), yaw (the rotary motion around the Y-axis) and roll (the rotary motion around the Z-axis).

Figure 5 presents the simulated deformations, the propagation of light, vibration amplitudes and the corresponding actuation characteristics (focal lengths, actuation displacements and rotation angles) for the PM-based ALENS. The output performance was measured as a function of the frequency (under 120 V/mm) and electric field (at 2 Hz). All the desired motion modes can generate steady, controllable, and repeatable outputs displacement in a wide frequency band from 1 Hz to 400 Hz, and the displacements increase linearly with the increase of electric field from 120 V/mm to 1600 V/mm. For the applied electric field of 1600 V/mm, the AF, AF*, the linear motions along the X- and Y-axis displacements of the PM-based ALENS are 69.8 μm, 31.4 μm, 4.88 μm and 5.05 μm, respectively. In addition, the hysteresis in the ALENS is the main factor that affects its displacement accuracy, which

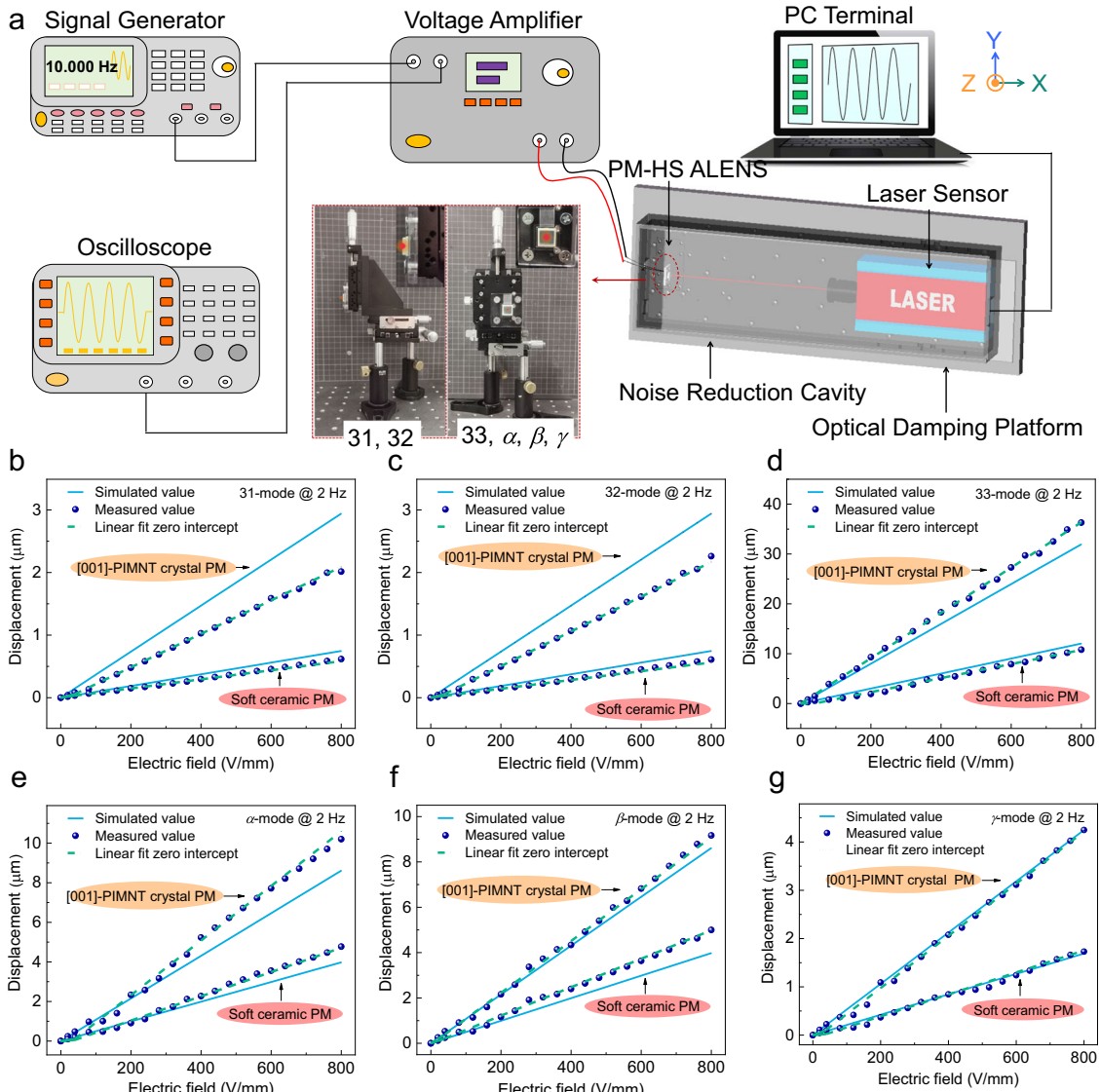

**Fig. 3 | Experimental verification of the PM with the vibration displacement responses. a** The experimental setup for measuring the output displacements of the PM. The simulated and experimental vibration displacements of the PM in (**b**) artificial 31-mode, (**c**) artificial 32-mode, (**d**) artificial 33-mode, (**e**) $\alpha$-mode, (**f**) $\beta$-mode and (**g**) $\gamma$-mode (with LTR = 40) under different electric fields (from 0 V/mm to 800 V/mm) at 2 Hz.

may affect the properties of the PM-based ALENS. Supplementary Figs. 8 and 9 depicts the hysteresis of the piezoelectric materials and the PM-based ALENS. The maximum hysteresis of the PM-based ALENS at 2 Hz, 10 Hz and 50 Hz are relatively low, being 3.9%, 3.8% and 4.2% respectively, which indicates that the PM-based ALENS exhibits high control accuracy and operation reliability. As shown in Supplementary Fig. 10, the impedance spectra have two obvious resonance peaks at 2.9 kHz and 13.8 kHz. When the working frequency is below 2 kHz, the output displacement of the PM-based ALENS for the AF remains basically stable. Thus, the minimum response time of the PM-based ALNES is around 0.5 ms according to the Supplementary Eq. (S7), which fulfills the requirement of optical systems (~10 ms). Moreover, the output displacement of the PM-based ALENS maintains almost unchanged over 14400 vibration cycles in the durability test, representing its long-term robustness, high reliability and good stability (Supplementary Fig. 11). The variation of displacement is only 0.8% over 500 s, and the creep characteristic of the PM-based ALENS is minimal (Supplementary Fig. 12). Detailed analysis can be found in Supplementary Note 2.

Considering the practical applications, we analyzed and calculated the actuation performance corresponding to the output displacements (Fig. 5). The focal lengths of the PM-based ALENS with respect to electric fields is determined by the thin-lens equation[23],

$$f = R/(n_l - 1) \tag{1}$$

where the refractive index of the silicone oil $n_l$ for the PM-based ALENS is 1.5[25]. $R$ is the radius of curvature for the spherical surface of the

**Table 1 | The simulated and experimental values of the strains $\varepsilon_j$, j = 1, 2 and 3, for the PM based on soft PZT ceramics[22] and [OO1]-poled PIMNT single crystals[21] under the electric field of 800 V/mm**

| Motion modes | Simulated piezoelectric strains (%) | | Experimental piezoelectric strains (%) | |
|---|---|---|---|---|
| | **PZT-5** | **PIMNT** | **PZT-5** | **PIMNT** |
| Artificial 31-mode | 0.008 | 0.030 | 0.006 | 0.022 |
| Artificial 32-mode | 0.008 | 0.030 | 0.006 | 0.023 |
| Artificial 33-mode | 0.240 | 0.666 | 0.182 | 0.756 |

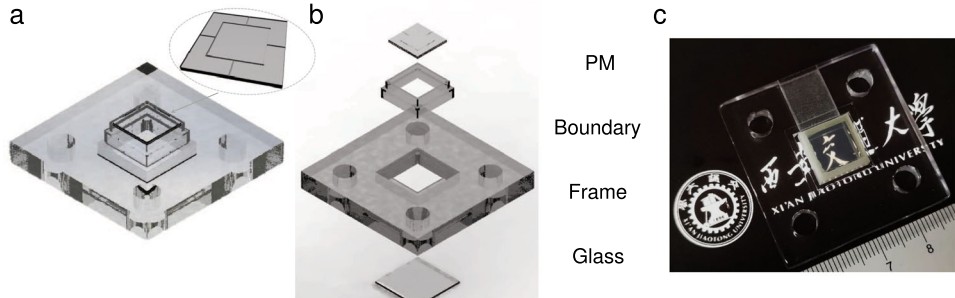

**Fig. 4 | Schematic and structure of the PM-based ALENS. a** The overall structure of the PM-based ALENS. (The inset at top right gives the structure of PM.) **b** Exploded figure of the PM-based ALENS. **c** The photograph of the PM-based ALENS prototype.

ALENS, determined by the Pythagorean proposition. As the applied electric field increases to 1600 V/mm, the focal lengths of AF and AF* modes declined from ∞ at 0 V/mm to 35.82 cm and 28.67 cm at 1600 V/mm, respectively. The actuation performance grows linearly as the electric field increases, in which the displacements along the X- and Y-axis, the tilt angles around the X-, Y- and Z-axis for the PM-based ALENS under 2 Hz and 1600 V/mm are 4.88 μm, 5.05 μm, 43.41′, 44.02′ and 17.90′, respectively. The device parameters of PM-based ALENS (including dimension, weight, applied electric field, working frequency, hysteresis properties, the displacements and the tilt angles) were comparable to those of the latest researches[13,26–29], as shown in Table 2. Meanwhile, it is worth noting that the most important advantage of the PM-based ALENS is simple structure, which simultaneously achieve AF and OIS functions by using only one PM.

### Applications for the PM-based ALENS

For the function verification test, the ray tracing, spot diagram and spot motion variation of the PM-based ALENS are investigated for the optical system (Fig. 6). Here, we demonstrated five representative functions, including AF, pitch, roll, the coupled modes of the AF and pitch, the coupled modes of the AF and roll.

For the AF function, the parallel light will focus via the PM-based ALENS at activated state. Simulation results show that the parallel beams will converge to the focal point, and the root mean square (RMS) radius ($r_{RMS}$) in spot diagram decreased from 1520 μm to 137 μm (Fig. 6a). Similarly, experimental results show that the spot diameter reduce from 780 μm at the non-focusing state to 240 μm at focusing states as the laser passes through the PM-based ALENS. As the PM-based ALENS generates the pitch, yaw and roll mode, light direction will be rotated around X-, Y- and Z-axis. For example, in the pitch mode, simulation results show that the parallel beams will be rotated around X-axis through the PM-based ALENS, and the spots were obviously deflected to the left or right along X-axis in the diagram. In addition, the laser spot periodically rotated around the X-axis, i.e., the center coordinates of the spot underwent reciprocating motions from (0, −0.205) to (0, 0.203), as illustrated in Fig. 6c.

Compared to the aforementioned basic mode, the PM-based ALENS can generate the coupled modes in which a convex lens was formed in the X-Y plane while producing axial translation or rotation along the X-, Y-axis or Z-axis. As expected, in the coupled modes of the AF and pitch, parallel beams were focused after passing through the PM-based ALENS while rotating around the X-axis, i.e., the center coordinates of the spot underwent reciprocating motions from (0, −0.135) to (0, 0.137) (Fig. 6d). In the coupled modes of AF and roll, parallel beams were focused after passing through the PM-based ALENS while moving clockwise around the Z-axis in circular motions (Fig. 6e). It is worth noting that the RMS radii of the coupled modes (-290 μm) were much smaller than that of the incident light (-1520 μm) in spot diagram, indicating that the modes were successfully coupled. As for the remaining operating modes, the

detailed information and the result analysis are summarized in Supplementary Note 3. All of the above-mentioned processes are shown in Supplementary Movie 2. The above results show that the PM-based ALENS can be applied to the AF and OIS functions of optical systems and miniaturization of multifunctional ALENS.

In summary, inspired by metamaterial design, we proposed a piezo metasurface to simultaneously achieve various motion modes with high strains. In this work, we fabricated a PM with ten pieces of PIMNT single crystal units, which can simultaneously achieve high piezoelectric strains ($\varepsilon_3 = 0.76\%$) and the desired motion modes (the linear motions along the X-, Y- and Z-axis, the rotary motions around the X-, Y- and Z-axis as well as the coupled modes) in a wide frequency range (from 1 Hz to 400 Hz). Based on the PM, we designed an ALENS that can simultaneously possess AF and OIS functions. Due to various motion modes with high strains for the PM, the PM-based ALENS can realize a wide range of focal length (35.82 cm to ∞) and the effective image stabilization with relatively large displacements (i.e., 5.05 μm along the Y-axis) and tilt angles (i.e., 44.02′ around the Y-axis). These characteristics suggest that our newly proposed PM is prospective for the design of advanced piezoelectric devices in industry.

## Methods
### FEM simulation of PM

The piezoelectric module in COMSOL Multiphysics was employed to perform all simulations using the finite element method (FEM), including the motion modes and output displacements. Materials parameters of the soft PZT5 ceramics were obtained from the built-in library, while that of the PIMNT crystals were selected in ref. 30. The structures, polarization directions, applied voltage signals and simulated deformations of the PM during the desired motion modes are shown in Fig. 1, Supplementary Figs. 1–4.

### Fabrication and measurement of PM

The PIMNT single crystals grown by Xi'an Jiaotong University using a modified Bridgman technique were selected[31,32]. The crystals were oriented by the X-ray diffraction method and the orientations were along [100], [010] and [001]. The crystals were cut and polished into 10 mm$^{Length}$ × 10 mm$^{Width}$ × 0.25 mm$^{Thickness}$, and ITO materials with a thickness of 200 nm were sputtered on the main surfaces as the transparent electrode by magnetron sputtered. For poling, the ferroelectric test system (TF Analyzer 2000E, aixACCT) was used to create a bipolar triangle wave electric field of 1 Hz and 10 cycles to the sample with a peak amplitude of 1 kV/mm. The light transmittance of the PIMNT single crystal was measured, and detailed analysis can be found in Supplementary Fig. 15.

The boundary structure and the frame were prepared by 3D printing. The Young's modulus for the boundary structure is 742 kPa, measured by universal material testing machine (AI-GS, Shimadzu, Japan). The structure of the proposed PM-based ALENS is shown in

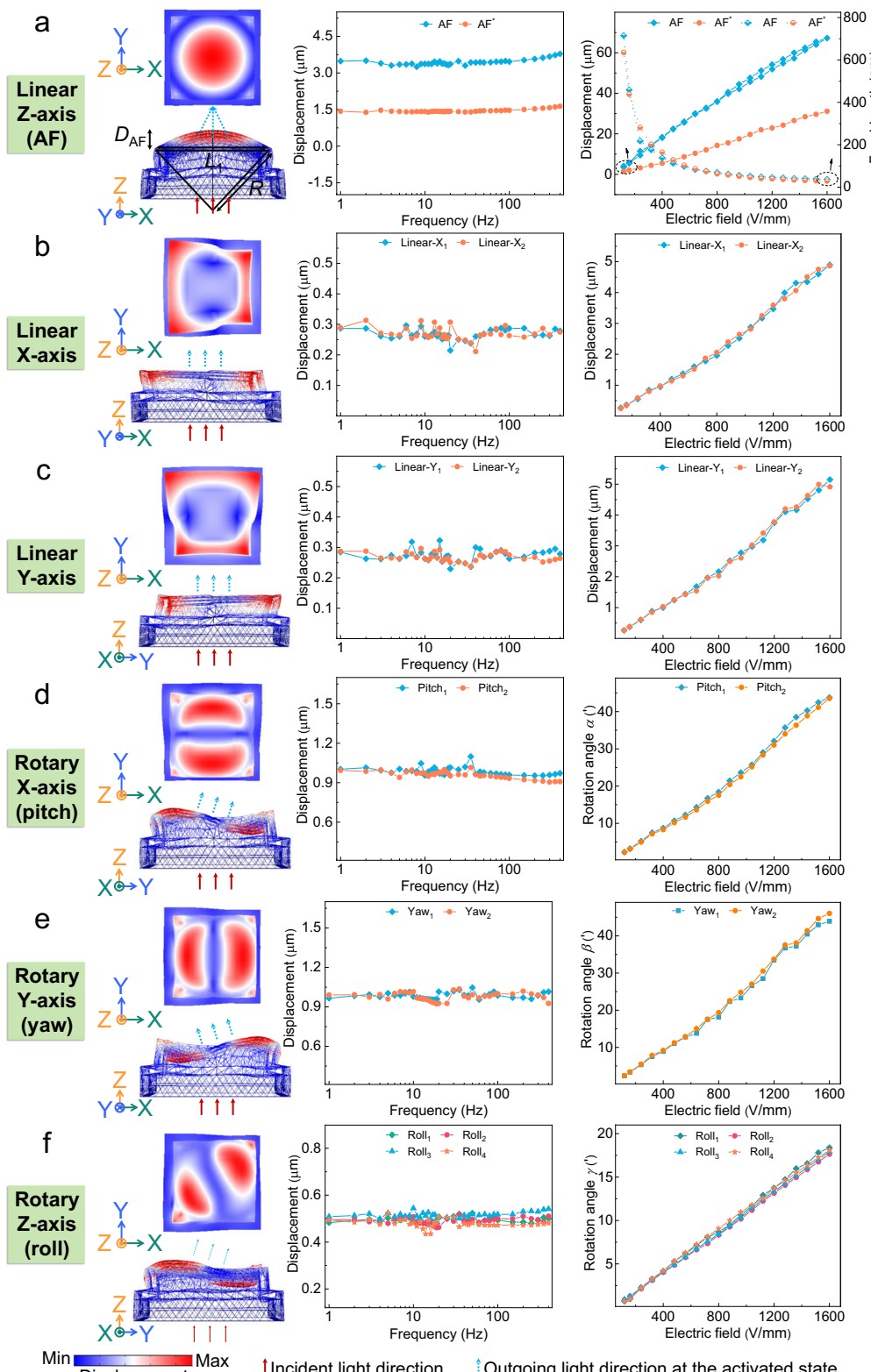

**Fig. 5 | The designed PM-based ALENS and corresponding experiments for vibration responses. a–f** Pictures on the left are the simulated deformations, including top view and sectional view; Pictures in the middle are vibration amplitudes under different frequencies (from 1 Hz to 400 Hz); Pictures on the right are actuation characteristics under different electric fields (from 120 V/mm to 1600 V/mm), including focal lengths, actuation displacements and rotation angles. **a** The linear motion along the Z-axis (AF), **b** the linear motion along the X-axis, **c** the linear motion along the Y-axis, **d** the rotary motion around the X-axis (pitch), **e** the rotary motion around the Y-axis (yaw) and **f** the rotary motion around the Z-axis (roll).

**Table 2 | Comparison of the PM-based ALENS with other devices**

| Actuation | Dimension (mm × mm) | Weight (g) | Working frequency (Hz) | Hysteresis (%) | Electric field (kV/cm) | Linear motions displacment (µm) | | | Rotary motions angle (') | | | References |
|---|---|---|---|---|---|---|---|---|---|---|---|---|
| | | | | | | X-axis | Y-axis | Z-axis | X-axis | Y-axis | Z-axis | |
| Piezoelectric | 13 × 13 | 9.4 | 1 ~ 2000 | 3.9 | 16 | 4.88 | 5.05 | 69.8 | 43.41 | 44.02 | 17.90 | This work |
| | 15 × 15 | – | 1 ~ 65,000 | – | 18 | 0.16 | 0.18 | 0.22 | 4 | 3.5 | – | Li et al.[13] |
| | 3 × 3 | – | 900 | – | 20 | 7.5 | 7.5 | 22 | 60 | 60 | 30 | Aktakka et al.[26] |
| | 15 × 15 | – | 1400 | – | – | 1.2 | – | 1.5 | – | – | – | Xu et al.[27] |
| Electromagnetic | φ3 | – | 4000 | – | – | 3.0 | 4.4 | 3.0 | 21.6 | 21.6 | 30 | Chen et al.[28] |
| Electrostatic | 8 × 8 | – | 500 | – | 17 | – | – | 18 | – | – | 103.2 | Mukhopadhyay et al.[29] |

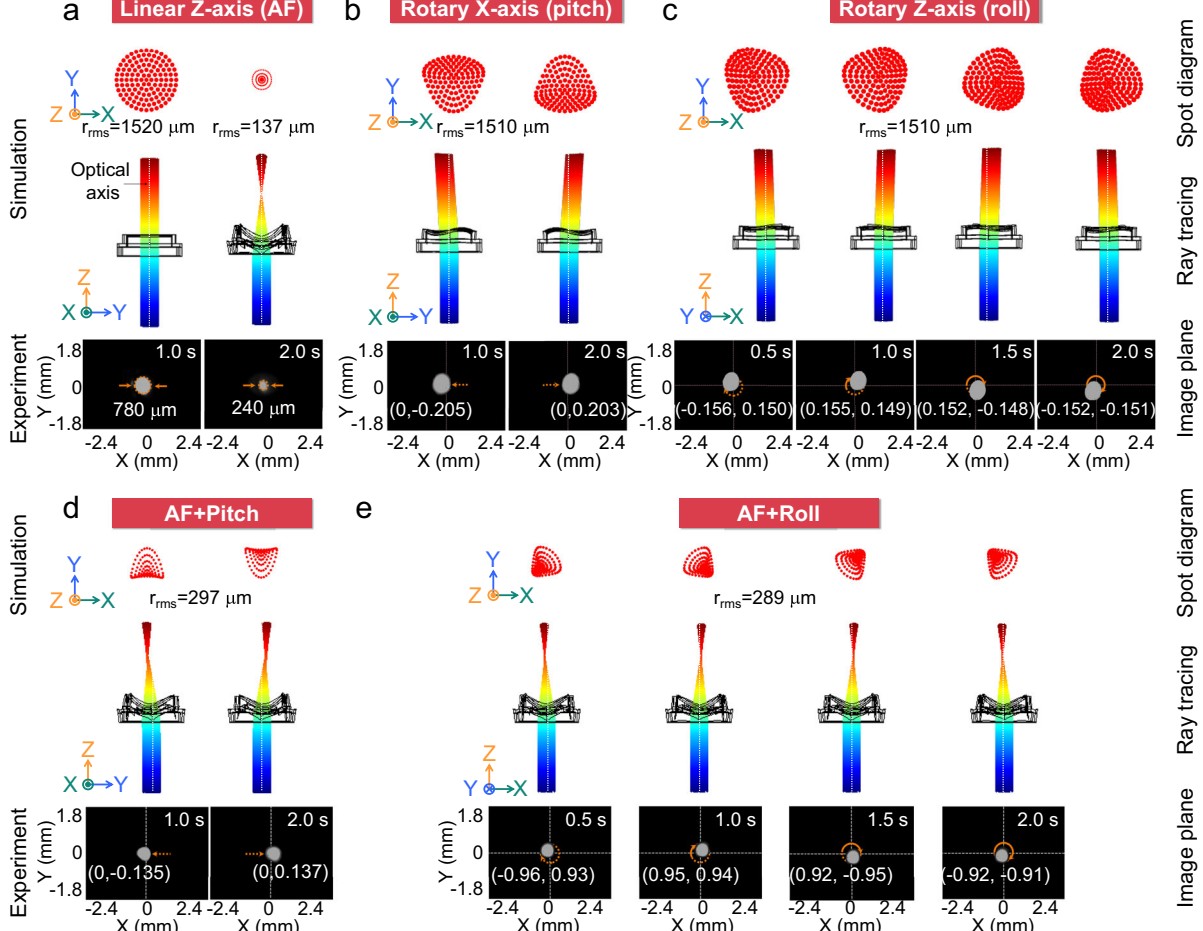

**Fig. 6 | The PM-based ALENS based on the AF and OIS functions under the basic modes and coupled modes for spot motion variation.** Ray optics simulations (including ray tracing and spot diagram) and dynamic characteristics experiment of optical spot for (**a**) AF, (**b**) pitch, (**c**) roll, (**d**) in the coupled modes of the AF and pitch, (**e**) in the coupled modes of the AF and roll in the PM-based ALENS.

Fig. 4, which consists of PM with [001]-poled PIMNT crystals, the boundary structure, a frame, a glass film and silicone oil.

The sinusoid wave and square wave applying signals were produced by a multi-function waveform generator (DG1022U, Rigol Technologies, China) after being outputted by a power amplifier (ATA-4052, Agitek, China). Then, the amplifying signals (1–400 Hz and 1600 V/mm) were applied for the PM-based ALENS and monitored continuously with an oscilloscope (DS1054, Rigol Technologies, China). All of the above displacements (including displacement vs frequency, displacement vs electric field) were tested using a laser rangefinder (LV-S01, Sunny Optical Instruments, China), as shown in Fig. 3a. Laser-produced light (HNLS-9-2.0, Thorlabs, USA) proceeded through the center of two neutral density filters, a pinhole, the PM-based ALENS and a CCD (MV-CA060-10GC, Hikvision, China), as shown in Supplementary Fig. 14.

### Reporting summary
Further information on research design is available in the Nature Portfolio Reporting Summary linked to this article.

## Data availability
The data that support the findings of this study are included with the manuscript as Supplementary Information. Any other relevant data are also available upon request from corresponding authors.

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

## Acknowledgements

This work was supported by National Natural Science Foundation of China (Grant Nos. 52325205, 52172129, and 52002312), Young Talent Fund of Association for Science and Technology in Shaanxi, China (Grant No. 20230419), and Natural Science Foundation of Shaanxi Province (Grant No. 2021GXLH-Z-025).

## Author contributions

The work was conceived and designed by L.Q., X.G., Z.X. and F.L.; L.Q. performed the finite element simulation and fabricated the samples and performed the experiments; L.Q. and X.G. performed the electro-mechanical properties measurement; K.R., C.Q. J.L., and H.J. assisted the properties measurements for the ALENS; F.L., X.G., and Z.X. super-vised the fabrication and test of the ALENS; L.Q., X.G., and F.L. drafted the manuscript; S.D. revised the manuscript; and all authors discussed the results.

## Competing interests

The authors declare no competing interests.
