## [Peer Review File · Nature Communications]

REVIEWER COMMENTS

Reviewer #1 (Remarks to the Author):

In the manuscript, the authors have achieved various types of ultrahigh strains in a wide frequency range by using a transparent piezo metasurface (PM), which consists of ten pieces of transparent PIMNT single crystal units, to produce ultrahigh strains and generate linear and rotation motions. Even though the presented results are technically sound, the topic is too specific. Additionally, the idea of using artificial structures to realize artificial vibration modes is not novel (see Ref. 13). I am not convinced that this contribution belongs to Nature Communications, as the physics involved is rather straightforward and there is nothing particularly surprising in the overall response. Anyway, I believe that the paper may belong to a more specialized journal.

Some comments the authors need to consider for the resubmission to other journals:

1. For piezoelectric units, they often have the hysteresis when the applied voltage is switched on and off. I wonder if the hysteresis has been observed in the experiment. Please add some discussions.
2. The actuation speed and endurance of the PM should be carefully investigated.
3. For the transparent material, the transmittance as a function of the wavelength should be measured.
4. The manuscript contains some grammatical errors that should be corrected. For example, it should be "piezo metasurface (PM)" instead of "piezo matasurface (PM)" in the abstract. It should be "a transparent piezo metasurfaces (PM)" instead of "a transparent piezo metasurface (PM)" on Page 4.

Reviewer #2 (Remarks to the Author):

The main contribution of this work can be concluded to propose a piezoelectric metasurface that can generate multiple motion modes (essentially similar to a specially designed piezoelectric actuator). The structural and electrical configurations are interesting and effective to produce multiple motions including linear motions of X-, Y- and Z-axis, rotary motions around X-, Y- and Z-axis, as well as their coupled motions. The application of the proposed device is also explored and discussed preliminarily. However, there are some problems in terms of concept, statement logic and research details. The experiments lack of more results to support the advancement of this article.

The more detailed comments are as follows:

1. The proposed structure is actually a piezoelectric actuator or manipulator, as it is used for optical beam manipulation in the experiments, but the author names it as a metasurface, which is worth reconsidering.
2. In the Para. 1 of Introduction, the author states “since piezoelectric materials possess a variety of vibration modes, piezoelectric devices are expected to simultaneously realize high-performance, multi-functions and miniaturization.” In fact, various vibration modes are the inherent dynamic results for all elastic mechanical structure, they are not the special properties of the piezoelectric materials.
3. In the Para. 2 of Introduction, the classic approach to construct multi-DOF piezoelectric motion system is using several multilayer piezoelectric stacks. Although this approach may lead to relatively complicated structures, the multi-DOF compound motions can be easily realized due to the independent control of each motion DOF, especially for serial configuration. While for the proposed structure in this work, the generation of multiple motion modes requires the excitation of multiple piezoelectric ceramic units. The excitation conditions of each motion mode are different with diverse voltage combinations and the function reuse exists in different piezoelectric ceramic units, which actually means that it is impossible to produce multiple independently controllable multi-DOF motions at the same time. Therefore, all statements about “simultaneously accomplishing various vibration modes” are false and misleading. The comparison statement between this work and the abovementioned classic approach using stack should be more rigorous. This is a crucial issue for the logic aspect.
4. The use of the concept of “vibration modes” is worth reconsidering due to the meaning of the vibration mode in this article is essentially indicating the motion modes. Actually, “vibration mode” is more used to describe the dynamic vibration situation of a structure, rather than the low-frequency quasi-static deformations. The concept of “motion mode” or “deformation mode” is recommended to replace the “vibration mode”.
5. Form the deformation working principle, the motion of each DOF can be stimulated independently, or the specific compound excitation of partial motions can be realized. However, the coupling effects among the various motion DOFs are not considered and presented. Furthermore, the random combinations of the six motions cannot be realized due to the function reuse of the piezoelectric ceramic unit.
6. The ultrahigh strain of the proposed structure is interesting, but the factors that contributed to this result are not discussed. Are they mainly special structure design or material particularity? Besides, what levels can be considered as high and ultrahigh strain are of concern. If there is no more clear data comparison, the statement about ultrahigh strain is inappropriate.
7. The author carefully studies the influences of the fixed supports on the response characteristics of the proposed structure, but the influences of the structure configurations of the PM itself are not discussed.
8. The Young’s modulus of the 3D printing fixed boundary is set as 750 kPa to ensure the high strain of the PM. A question is that the 3D printed prototype is not tested or evaluated to show actual result of its Young’s modulus in the experiment. This is necessary to check the consistency of simulation and experimental results.
9. The statement about the “boundary condition” and “boundary structure”, as well as “PM” and “PM boundary” should be clarified to avoid confusion in the part of simulation results.

10. As for the experimental results shown in Fig. 2e-h, why the length to thickness ratio is used, rather than the direct length or thickness parameters?
11. The size definition of the thickness of boundary structure is not consistent with its value presented in the experimental results.
12. The used excitation voltages in the simulation (400V/1mm or 100 V DC) and that in the experiment (400Vpp shown in Fig. 5) are not consistent. They should be modified to be uniform.
13. The dynamic characteristics of the PM are not provided and discussed in this article. As for an actuation device with multiple DOFs, the dynamic characteristics of each DOF are of concern to determine the practical feasible working condition. For example, the resonant frequency of PM in terms of diverse motion DOFs. The frequency-amplitude characteristics are recommended to test and add into the experiment.
14. The application of the PM-based ALENS is presented in the experiment to show the capability of focusing and imaging adjustment. But the presented results are focal length adjustments or image adjustments, not real autofocus operation. Autofocus usually means automatic operation processing. This point should be modified to be more rigorous.
15. The nonlinear response characteristics of the prototype are all not considered in this work. As motion devices with multiple motion modes working with a frequency range of 1Hz to 400Hz, the nonlinear hysteresis (in full frequency range) and creep (at low frequency) are two main characteristics affecting the service ability of the PM. Furthermore, the response characteristics of lower than 1Hz and over than 400Hz are also worth considering.
16. The ultrahigh strain means that the PM holds low stiffness performance. A question is that the continuous dynamic excitation will cause fatigue in the structure, which may lead to structure damage and further affect the lifetime. How to evaluate the lifetime and characteristic evolution of the PM?
17. As for the six motion modes of the PM, the positions of generating maximum deformation and node displacement are different. This is unlike other piezoelectric actuation devices that usually have a specific motion output end. In addition, the small size and multiple motion modes of the PM limit the possibility of installing additional execution ends, which seriously limits the application of the PM. These issues are not considered in the article.
18. In the method part, how to determine the position of generating maximum displacement and position size is not clarified.
19. In terms of the experimental condition and setting, the displacement measuring accuracy of the used laser displacement measuring device (LV_S01, a single-point laser vibration testing instrument) is not enough to test the small displacement of the PM in the specific position due to limitation of the device principle. The specialized laser displacement sensor with high measuring accuracy and bandwidth is recommended to continuously measure response characteristic of the PM, which benefits for showing the full characteristics. Furthermore, the measuring interval of the response under different frequency needs to shorten and the measuring range needs to extend.
20. The experimental results shown in Fig. 3 need to cover wider excitation voltage range, i.e., 0V to 10V and over 100V.

21. From the results shown in Fig.4a, the PM with electrodes seems to be non-transparent. How does the laser beam penetrate it in the application experiments?
22. As for the experiment result shown Fig. 5d-f, using the unit of “ μm ” to describe the rotary motions is inappropriate. The angular unit is recommended to show the rotary displacement directly.
23. The more experimental results and discussion to further demonstrate the application value (the capability to solving practical problem) of the PM manipulating the laser beam need to add into the experiments.
24. The characteristic comparison listed in the Table 2 is insufficient. The more details about the size, weight, motion end, etc. should be supplemented and discussed.
25. As shown in the Supplementary Fig. 2, the PM is installed in the different boundary structure, a question is how to ensure the installing consistency under multiple operations, which may lead to different fixing stiffness and further affect experimental comparison results.

Reviewer #3 (Remarks to the Author):

This article proposed piezoelectric metasurface to simultaneously achieve various vibration modes with ultrahigh piezoelectric strains. They fabricated a piezo metasurface with ten pieces of PIMNT single crystals units, which can simultaneously achieve desired vibration modes and ultrahigh piezoelectric strains in a frequency range of 1-400 Hz. Using piezo metasurface, the authors also designed PM adaptive lens (ALENS) that can simultaneously possess autofocus (AF) and optical image stabilization (OIS) functions. The subject is original and interesting, and it is rich in technical details. The paper can be considered for possible publication subject to the following comments.

Following are some comments and suggestions:

- Page 4, Line 2, ref 13 should be named and correctly cited.
- Page 5, line 7-11, “With applying the AC voltages, the specific region of the PM can generate transverse-extension deformation or axial-bending deformation. Based on the synergistic strain effect among the piezoelectric units^{13,14}, the PM can generate greatly enhanced deformation along a specific direction to achieve the desired vibration modes”, did authors considered structural nonlinearity under such deformation?
- Page 6, para 2 and 3, and page 7, para 2, have identical sentences and seems more a repetition.
- Page 8, last two lines, “It should be noted that if the Young's modulus the boundary is too low it cannot bear the whole weight of the PM”. This sentence is not clear and needs to be revised.

- Page 10, line 9-11, “The output displacements remarkably correlated linearly with the operating voltages at 2 Hz in the range of 10 ~ 100 Vpp, which are in good accordance with the simulation data”, authors are suggested to explain this sentence. How the output displacement correlated linearly with applied voltage?
- The practical significance of this study is unclear. Authors just mentioned that “This research may inspire miniaturization and integration for multi-degrees-of-freedom systems”. This is not enough to catch the interest of industry or applied engineering research community.

Responses to Reviewers' comments

Reviewers' comments and authors' replies:

[Reviewers' comments are in black; Author responses are in blue; Revisions in the manuscript are highlighted.]

Response to reviewer #1

Comment: In the manuscript, the authors have achieved various types of ultrahigh strains in a wide frequency range by using a transparent piezo metasurface (PM), which consists of ten pieces of transparent PIMNT single crystal units, to produce ultrahigh strains and generate linear and rotation motions. Even though the presented results are technically sound, the topic is too specific. Additionally, the idea of using artificial structures to realize artificial vibration modes is not novel (see Ref. 13). I am not convinced that this contribution belongs to Nature Communications, as the physics involved is rather straightforward and there is nothing particularly surprising in the overall response. Anyway, I believe that the paper may belong to a more specialized journal. Some comments the authors need to consider for the resubmission to other journals:

Reply: Thanks for the comment and also thanks for the recognition of our presented results. In our present work, we proposed a new design principle for the piezo metasurface (PM), and verify its functions with the application of an adaptive lens (ALENS). Meanwhile, we are aware of the contribution of Dong *et al.* (Ref. 13) on the design of piezoelectric metamaterials for generating basic modes. However, the proposed principle of piezo metasurfaces and the electromechanical devices designed in our present work are very different from those reported in previous works. The novelty and advances of our work are listed as follows.

(1) We designed the PM based on the uniform boundary condition of a topological structure, while the boundary conditions are not totally unified in previous works. For example, in Ref. 13, the artificial torsion mode around 1-axis and 2-axis were achieved under fixed constrains in the planes normal to 1- and 2-axis, respectively. This feature of our present work, i.e., the unified boundary structure, can lead to a broader range of electromechanical devices that serve multiple functions from various motion modes.

(2) The PM reported in our work is totally different from Ref. 13 in terms of the unit ordered arrangement. For example, our proposed PM is arranged in X-Y plane, e.g., 5×2 or 9×2 arrayed structure, where the units are ordered with one in the center while others are arranged with center outward expansion method. As shown in Fig. R1, the center outward expansion method is an orderly structure construction method in which piezoelectric units expand outward circle by circle from the center of the structure with a consideration of required motion modes. Specifically, A_{10} (A_{20}) unit is in the center, A_{11} , A_{12} , A_{13} , and A_{14} (A_{21} , A_{22} , A_{23} and A_{24}) units are the first circle, and A_{15} , A_{16} , A_{17} and A_{18} (A_{25} , A_{26} , A_{27} and A_{28}) units are the second circle, and it is continuously expanded outwards along this way (Fig. R1). However, the piezoelectric metamaterial reported by Ref. 13 is fabricated with an $(2 \times 2 \times 2)$ arrayed structure, where the piezo strain units were simply stacked together.

(3) The main achievements of our work also include the realization of ALENS with both adjusting focus (AF) and optical image stabilization (OIS) functions in a single prototype, which are not involved in previous literature. In our present work, the PM were successfully utilized to design ALENS, and the newly fabricated ALENS with both AF and OIS functions is demonstrated for the very first time. Owing to the high strain of the desired linear motions and rotary motions, our fabricated ALENS can achieve a relatively wide range of focal length (35.82 cm to ∞) and an effective image stabilization with relatively large displacements (i.e., $5.05 \mu\text{m}$ along the Y-axis) and tilt angles (i.e., $44.02'$ around the Y-axis).

Fig. R1. Schematic diagram of the center outward expansion method. (This figure has been added as Supplementary Fig. 1 in the revised manuscript.)

Comment: 1) For piezoelectric units, they often have the hysteresis when the applied voltage is switched on and off. I wonder if the hysteresis has been observed in the experiment. Please add some discussions.

Reply: We appreciate the very valuable comment. We will discuss the hysteresis from two aspects.

(1) The piezoelectric materials for PM

Compared with the commercial PZT5 ceramics (~20%), the $\text{Pb}(\text{In}_{1/2}\text{Nb}_{1/2})\text{O}_3\text{-Pb}(\text{Mg}_{1/3}\text{Nb}_{2/3})\text{O}_3\text{-PbTiO}_3$ (PIMNT) single crystals have the advantage of very low strain hysteresis (~3%), reducing complexity and cost to the feedback control of piezoelectric devices (Fig. R2a).

(2) The structure of PM-based ALENS

As shown in Fig. R2b, the hysteresis in the ALENS was measured. The calculated results show that the maximum hysteresis is as low as 3.9%, exhibiting high control accuracy and high operation reliability. The low displacement hysteresis of the PM-based ALENS was mainly associated with the strain hysteresis of the PIMNT single crystals.

Fig. R2. The hysteresis characteristic. (a) Unipolar strain curves for the soft PZT5 ceramics and the PIMNT single crystals under 20 kV/cm at 1 Hz. (b) The linear motion along the Z-axis for the designed PM-based ALENS. (This figure has been added as Supplementary Fig. 7 in the revised manuscript.)

According to this comment, we added the following description in the revised manuscript:

“In addition, the hysteresis in the ALENS is the main factor that affects its displacement accuracy,

which may affect the properties of the PM-based ALENS. **Supplementary Fig. 8** depicts the hysteresis of the piezoelectric materials and the PM-based ALENS. The maximum hysteresis of the PM-based ALENS is as low as 3.9%, exhibiting high control accuracy and high operation reliability.”

Comment: 2) The actuation speed and endurance of the PM should be carefully investigated.

Reply: We thank the reviewer for the valuable comment. Here, we understand the actuation speed by the reviewer as the response speed of the PM. In the following, we would like to discuss the response speed and endurance of the PM-based ALNES.

(1) The response speed of the PM-based ALNES

The response speed is the degree of rapidity with which the ALENS transitions from one state to another after a given operation [Abe K., *et al.* Jpn. J. Appl. Phys. 21.7A, L408 (1982)], [Lou, Zheng, *et al.* Nano Energy 23, 7-14 (2016)], which can be represented by the response time. It is calculated as follows [Sun C. L., *et al.* Applied Physics A-Mater. 84, 385-389 (2014)],

$$t_{\text{response time}} = \frac{1}{f_{\text{operation frequency}}} \quad (1)$$

According to the above formula, the response time is closely related to the frequency of the operating electric field. As shown in Fig. R3a, the impedance spectra of PM-based ALNES have two obvious resonance peaks at 2.9 kHz and 13.8 kHz. In our present work, the PM-based ALNES works under the quasi-static condition, so its operating frequency is much lower than the resonance frequency. In order to acquire the minimum response time, we tested and added the output displacement of the PM-based ALENS under different frequencies into the experiment part. As shown in Fig. R3b, when the working frequency is below 2 kHz, the output displacement of the PM-based ALENS for the AF remains basically stable. Thus, the minimum response time of the PM-based ALNES is around 0.5 ms according to the Equation (1), which fulfills the requirement of optical systems (~ ms).

Fig. R3. The frequency characteristic. (a) Impedance and phase spectra of the PM-based ALENS as a function of frequency ranging from 20 Hz to 30 kHz at room temperature. (b) Vibration displacement amplitudes of the PM-based ALENS for the AF under different frequencies. (This figure has been added as Supplementary Fig. 8 in the revised manuscript.)

(2) The endurance of the PM-based ALENS

The endurance characteristics of the PM-based ALENS was measured at 2 Hz under 1600 V/mm, as shown in Fig. R4. The inset figures show transient response of the displacement for the PM-based ALENS, where the output displacements exhibit good repeatability under 1600 V/mm over 14400 vibration cycles, representing its long-term robustness, high reliability and good stability.

Fig. R4. Endurance characteristics of the PM-based ALENS (The inset shows the transient surface displacement of the PM-based ALENS, and relative variation of displacement is calculated via the ratio of output displacement

to max output displacement). (This figure has been added as Supplementary Fig. 9 in the revised manuscript.)

According to this comment, we added the following description the revised manuscript:

“As shown in **Supplementary Fig. 9**, the impedance spectra have two obvious resonance peaks at 2.9 kHz and 13.8 kHz. When the working frequency is below 2 kHz, the output displacement of the PM-based ALENS for the AF remains basically stable. Thus, the minimum response time of the PM-based ALNES is around 0.5 ms according to the **Equation (S7)**, which fulfills the requirement of optical systems (~ 10 ms).”

“Moreover, the output displacement of the PM-based ALENS maintains almost unchanged over 14400 vibration cycles in the durability test, representing its long-term robustness, high reliability and good stability (**Supplementary Fig. 10**). Detailed analysis can be found in **Supplementary Note 2**.”

Comment: 3) For the transparent material, the transmittance as a function of the wavelength should be measured.

Reply: We thank the reviewer for the valuable comment. As shown in Fig. R5a, the PIMNT single crystal is clearly transparent. The light transmittance of the PIMNT single crystal is found to be very close to its theoretical limit (i.e., optical loss is only from surface reflection). The optical transmittance of transparent PIMNT crystals is about 70% in the wavelength range of 550 to 2000 nm (Fig. R5b), which is close to lithium niobate single crystal (LiNbO₃, LN) [Ajay, D. *et al.* J. Appl. Phys. 68, 5804-5809 (1990)]. In addition, we can greatly improve the crystal transmittance through the coating of antireflective film on the crystal surfaces. In particular, at a wavelength of 1064 nm the optical transmittance of PIMNT crystal can be enhanced up to 99.6%.

Fig. R5. (a) Photograph of transparent PIMNT crystal. (b) Light transmittance of transparent PIMNT crystals with and without antireflection film. (This figure has been added as Supplementary Fig. 14 in the revised manuscript.)

According to this comment, we added the following description the revised manuscript:

“The light transmittance of the PIMNT single crystal was measured, and detailed analysis can be found in **Supplementary Fig. 14.**”

Comment: 4) The manuscript contains some grammatical errors that should be corrected. For example, it should be "piezo metasurface (PM)" instead of "piezo matasurface (PM)" in the abstract. It should be “a transparent piezo metasurfaces (PM)” instead of “a transparent piezo metasurface (PM)” on Page 4.

Reply: Thanks very much for pointing out the grammatical errors in our manuscript. We have carefully proofread our manuscript and revised these errors.

Response to reviewer #2

Comment: The main contribution of this work can be concluded to propose a piezoelectric metasurface that can generate multiple motion modes (essentially similar to a specially designed piezoelectric actuator). The structural and electrical configurations are interesting and effective to produce multiple motions including linear motions of X-, Y- and Z-axis, rotary motions around X-, Y- and Z-axis, as well as their coupled motions. The application of the proposed device is also explored and discussed preliminarily. However, there are some problems in terms of concept, statement logic and research details. The experiments lack of more results to support the advancement of this article.

Reply: We greatly appreciate the reviewer for considering our work interesting. According to the reviewer's comments, we have further revised our manuscript and added more results to elaborate our work.

Comment: 1) The proposed structure is actually a piezoelectric actuator or manipulator, as it is used for optical beam manipulation in the experiments, but the author names it as a metasurface, which is worth reconsidering.

Reply: Thanks for the comment. Following the definition of electromagnetic and optical metasurfaces [Capolino Filippo, *et al.* Appl. Phys. Applied Physics Lett. 120, 060401 (2022); Hu Jie, *et al.* Frontiers in Physics Front Phys-Lausanne 8, 586087 (2021)], we determined the piezo metasurfaces (PM) as the two-dimensional piezoelectric materials composed of periodic piezoelectric units, such as an (5×2 or 9×2 or 21×2) arrayed structure. Due to its unique structure, PM is able to generate multiple motion modes via applying electric fields. The distinctive feature of our designed PM is the multidimensional motions on the surface by controlling different regions of the metasurfaces to produce the desired motion modes. Thus, the PM is quite different from the traditional actuator, which generally possess only one motion output end. For example, by using only one PM, AF and OIS functions can be simultaneously realized. Obviously, these functions cannot be achieved by a traditional actuator.

According to this comment, we added a sentence in the revised manuscript to clarify the definition of piezo metasurfaces:

“Piezo metasurface is a two-dimensional piezoelectric material consisting of periodic piezoelectric

units, which generates multiple motion modes by applying electric fields.”

Comment: 2) In the Para. 1 of Introduction, the author states “since piezoelectric materials possess a variety of vibration modes, piezoelectric devices are expected to simultaneously realize high-performance, multi-functions and miniaturization.” In fact, various vibration modes are the inherent dynamic results for all elastic mechanical structure, they are not the special properties of the piezoelectric materials.

Reply: We thank the reviewer for the valuable comment. This statement is indeed misleading and we have made modifications in the revised manuscript, as shown in below.

“Under the external electric field, piezoelectric materials can generate stresses and strains through inverse piezoelectric effect, which can produce a variety of vibration modes.”

Comment: 3) In the Para. 2 of Introduction, the classic approach to construct multi-DOF piezoelectric motion system is using several multilayer piezoelectric stacks. Although this approach may lead to relatively complicated structures, the multi-DOF compound motions can be easily realized due to the independent control of each motion DOF, especially for serial configuration. While for the proposed structure in this work, the generation of multiple motion modes requires the excitation of multiple piezoelectric ceramic units. The excitation conditions of each motion mode are different with diverse voltage combinations and the function reuse exists in different piezoelectric ceramic units, which actually means that it is impossible to produce multiple independently controllable multi-DOF motions at the same time. Therefore, all statements about “simultaneously accomplishing various vibration modes” are false and misleading. The comparison statement between this work and the abovementioned classic approach using stack should be more rigorous. This is a crucial issue for the logic aspect.

Reply: We appreciate the very valuable comment. Our designed arrayed PM has the ability to simultaneously accomplish multiple motion modes. The detailed method is clarified as follows.

(1) The method for simultaneously accomplishing two motion modes

As shown in Fig. R6, the piezoelectric units are divided into two groups, where A_{10} and A_{20} are classified as group I, where the A_{11} , A_{12} , A_{13} , A_{14} , A_{21} , A_{22} , A_{23} and A_{24} are classified as group II. By applying predetermined electric fields, group I realizes the linear motion along the Z-axis, and group

II can realize the linear motions along the X-, Y- and Z-axis, and/or the rotary motions around the X-, Y- and Z-axis. The two motion modes can be independently excited and thus two motion modes can be simultaneously accomplished.

For example, for simultaneously accomplishing the linear motion along the Z-axis and the linear motion along the X-axis (Fig. R6a), the A_{20} with voltage signal V_5 and the A_{10} with voltage signal V_6 are classified as group I, while the A_{11} , A_{12} , A_{21} and A_{22} units with voltage signal V_1 and the A_{13} , A_{14} , A_{23} and A_{24} units with voltage signal V_2 are classified as group II. In this case, various motion modes can be produced for group I and II, i.e., group I generates the arching (or incurvaing) deformation along the positive (or negative) direction of the Z-axis, group II causes the whole PM by the lateral translation deformation along the X-axis. Therefore, simultaneously accomplishing the linear motion along the Z-axis and the linear motion along the X-axis can be realized.

(2) The method for simultaneously accomplishing three motion modes

As shown in Fig. R7, the piezoelectric units of an 9×2 arrayed PM are divided into three groups, where A_{10} and A_{20} are classified as group I, the A_{11} , A_{12} , A_{13} , A_{14} , A_{21} , A_{22} , A_{23} and A_{24} are classified as group II, and the A_{15} , A_{16} , A_{17} , A_{18} , A_{25} , A_{26} , A_{27} and A_{28} are classified as group III. By applying predetermined electric fields, group I realizes the linear motion along the Z-axis, group II and group III can realize the linear motions along the X-, Y- and Z-axis, and/or the rotary motions around the X-, Y- and Z-axis. The three motion modes can be independently excited, so one or several motion modes can be simultaneously accomplished by freely matching. The applied voltage signals and simulated deformations of simultaneously accomplish three multiple motion modes is shown in Fig. R7.

(3) The method for simultaneously accomplishing six motion modes

As shown in Fig. R8, six motion modes can be simultaneously realized by more ordered units with center outward expansion method (an 21×2 arrayed PM).

According to this comment, we have added the above contents to Supplementary materials (Supplementary Note 1) in the revised manuscript.

Fig. R6. Design of an (5 × 2) arrayed PM to simultaneously accomplish two motion modes. (a-b) The applying voltage signals and simulated deformations of simultaneously accomplish (a) the linear motion along X-axis and the linear motion along Z-axis, (b) the linear motion along Y-axis and the linear motion along Z-axis, (c) the linear motion along Z-axis and the rotary motion around X-axis, (d) the linear motion along Z-axis and the rotary motion around Y-axis, (e) the linear motion along Z-axis and the rotary motion around Z-axis in the PM. (This figure has been added as Supplementary Fig. 2 in the revised manuscript.)

Fig. R7. Design of an (9×2) arrayed PM to simultaneously accomplish three motion modes. (a) Schematic structure of an (9×2) arrayed PM. (b-d) The applied electric voltage signals and deformation schematics of simultaneously accomplish three motion modes. (b) the linear motion along the X-axis and the linear motion along the Y-axis, (c) the linear motion along the X-axis and the rotary motion along the X-axis, (d) the linear motion along the X-axis, the linear motion along the Y-axis and the linear motion along the Z-axis. (This figure has been added as Supplementary Fig. 3 in the revised manuscript.)

Fig. R8. Design of an (21×2) arrayed PM to simultaneously accomplish six motion modes. (a) Schematic structure of an (21×2) arrayed PM. (b) The deformation schematics of simultaneously accomplish six motion modes. (This figure has been added as Supplementary Fig. 4 in the revised manuscript.)

Comment: 4) The use of the concept of “vibration modes” is worth reconsidering due to the meaning of the vibration mode in this article is essentially indicating the motion modes. Actually, “vibration mode” is more used to describe the dynamic vibration situation of a structure, rather than the low-frequency quasi-static deformations. The concept of “motion mode” or “deformation mode” is recommended to replace the “vibration mode”.

Reply: Thanks for the very constructive comment. Following the suggestion, we have replaced

“vibration mode” by “motion mode”.

Comment: 5) Form the deformation working principle, the motion of each DOF can be stimulated independently, or the specific compound excitation of partial motions can be realized. However, the coupling effects among the various motion DOFs are not considered and presented. Furthermore, the random combinations of the six motions cannot be realized due to the function reuse of the piezoelectric ceramic unit.

Reply: Thanks for the valuable comment. Because the designed PM worked under the quasi-static condition, there is no coupling effect among various motion modes.

In addition, it is possible to realize a random combination of the six motions by our designed PM, as discussed below.

As shown in Fig. R8, the piezoelectric units are divided into six groups, where A_{10} and A_{20} are classified as group I, the A_{11} , A_{12} , A_{13} , A_{14} , A_{21} , A_{22} , A_{23} and A_{24} are classified as group II, the A_{15} , A_{16} , A_{17} , A_{18} , A_{25} , A_{26} , A_{27} and A_{28} are classified as group III, the A_{19} , $A_{1(10)}$, $A_{1(11)}$, $A_{1(12)}$, A_{29} , $A_{2(10)}$, $A_{2(11)}$ and $A_{2(12)}$ are classified as group IV, the $A_{1(13)}$, $A_{1(14)}$, $A_{1(15)}$, $A_{1(16)}$, $A_{2(13)}$, $A_{2(14)}$, $A_{2(15)}$ and $A_{2(16)}$ are classified as group V, and the $A_{1(17)}$, $A_{1(18)}$, $A_{1(19)}$, $A_{1(20)}$, $A_{2(17)}$, $A_{2(18)}$, $A_{2(19)}$ and $A_{2(20)}$ are classified as group VI. By applying predetermined electric fields, group I realizes the linear motion along the Z-axis, group II-V can realize the linear motions along the X-, Y- and Z-axis, and/or the rotary motions around the X-, Y- and Z-axis. The six motion modes can be independently excited, so one or several motion modes can be simultaneously accomplished by freely matching.

As shown in Fig. R8, six motion modes can be simultaneously realized by more ordered units with the center outward expansion method (an 21×2 arrayed PM).

In conclusion, a combination of the six motions can be realized via the PM. According to this comment, we have added the above contents to Supplementary materials (Supplementary Note 1).

Comment: 6) The ultrahigh strain of the proposed structure is interesting, but the factors that contributed to this result are not discussed. Are they mainly special structure design or material particularity? Besides, what levels can be considered as high and ultrahigh strain are of concern. If there is no more clear data comparison, the statement about ultrahigh strain is inappropriate.

Reply: Thanks for the very constructive comment. The high output strain of the PM involves two

factors: the intrinsic properties of piezoelectric materials and boundary structure.

(1) Intrinsic properties of piezoelectric materials. As shown in Fig. R2a, the strain of the PIMNT single crystal is 0.27% at an electric field of 20 kV/cm, which is double the value of the soft PZT5 ceramics (~0.14%). Thus, in this work, we selected PIMNT as the piezoelectric elements to achieve high output strains.

(2) Boundary structure. The output performance (the piezoelectric strain ε_3 of the 33-mode) for the PM can be also optimized in terms of the Young's modulus and the boundary structure (Fig. R9a-d).

(i) The output strains ε_3 of the 33-mode with respect to the Young's modulus of the boundary are represented in Fig. R9a. As expected, the ε_3 values increased as the Young's modulus decreased. In our design, the Young's modulus of the boundary is set to be 750 kPa. (ii) The relationship of boundary structure and the output displacement is presented in Fig. R9b. Compared to the other boundary structures, the PM₂ possess the highest output strain (0.405%) because the stress distribution of the PM₂ tends to be uniform and stays at a relatively high level after optimization.

Additionally, the maximum electric-field-induced strain of piezoelectric ceramics is around 0.25%, thus we consider the strain of PM (0.76%) to be large. We would like to note here that there is no specific criterion for ultrahigh strain, thus we respect the reviewer's comment and replaced "ultrahigh strain" by "high strain" in the revised manuscript.

Fig. R9. Output performance for the desired motion modes as a function of the boundary structure and geometric dimensions of the PM by FEM simulation. (a) The strain ε_3 variation of the PM as a function of the

Young's modulus of the boundary. (b) The strain ε_3 variation of 33-mode with the boundary structure. (The insets show the boundary structures of PM₁ and PM₂.) (c) The strain ε_3 variation of 33-mode with the geometric dimensions of the boundary structure. (d) Photograph of the boundary structure. (e, f) The strain (ε_j , $j = 1$ and 3) variation of (e) 31-mode and (f) 33-mode with the geometric dimensions of the piezoelectric units for the PM. (g, h) The rotation angle (α and γ) variation of (g) α -mode and (h) γ -mode with the geometric dimensions of the piezoelectric units for the PM. The insets show the geometrical diagrams and deformations of the simulated motion modes by using PM₂. The black arrows represent displacement directions. The electric field of 400 V/mm was applied on the soft PZT ceramics and [001]-poled PIMNT single crystals. (Fig. 2 in the revised manuscript has been updated.)

Comment: 7) The author carefully studies the influences of the fixed supports on the response characteristics of the proposed structure, but the influences of the structure configurations of the PM itself are not discussed.

Reply: Thanks for the valuable comment. In this work, our main efforts were focused on the PM whose overall structure presents a square shape. We fully agree with the reviewer that it is worthy to further investigate influences of structure configurations on the response of PM. According to this comment, we will put more attention on the design of the overall structure of the PM in our future study.

Comment: 8) The Young's modulus of the 3D printing fixed boundary is set as 750 kPa to ensure the high strain of the PM. A question is that the 3D printed prototype is not tested or evaluated to show actual result of its Young's modulus in the experiment. This is necessary to check the consistency of simulation and experimental results.

Reply: Thanks for the question. We have added the Young's modulus of the 3D printing fixed boundary in the experimental section. The Young's modulus of the 3D printing fixed boundary is measured to be 742 kPa by universal material testing machine (AI-GS, Shimadzu, Japan), as shown in Fig. R10.

Fig. R10. The Young's modulus of the 3D printing fixed boundary in the experiment by universal material testing machine.

According to this comment, we added the following description in the revised manuscript:

“The Young's modulus for the boundary structure is 742 kPa, measured by universal material testing machine (AI-GS, Shimadzu, Germany).”

Comment: 9) The statement about the “boundary condition” and “boundary structure”, as well as “PM” and “PM boundary” should be clarified to avoid confusion in the part of simulation results.

Reply: Thanks for the comment. As suggested by the reviewer, we have revised this content. To avoid confusion, we uniformly expressed as boundary structure.

Comment: 10) As for the experimental results shown in Fig. 2e-h, why the length to thickness ratio is used, rather than the direct length or thickness parameters?

Reply: We thank the reviewer for the very valuable comment. At the length-to-thickness ratio (LTR) equal to 40, we analyzed the output performance of the PM with different sizes (including the length and thickness of the PM) by FEM simulation. As shown in Fig. R11, the piezoelectric strain of the PM reaches stabilization after a small decrease as the length of the PM increases. According to a comparison of PMs with various lengths and thicknesses, we confirmed that the length-to-thickness ratio is a key parameter for the output strain of PM. Thus, we choose the length to thickness ratio to optimize output performance.

Fig. R11. The output performance of the PM with the various length of the piezoelectric material under the LTR of 40 by FEM simulation.

Comment: 11) The size definition of the thickness of boundary structure is not consistent with its value presented in the experimental results.

Reply: Thanks for the comment. The thickness of boundary structure is marked in Fig. R12, where Fig. R12a presents the definition of the thickness of boundary structure and Fig. R12b gives the picture of ALENS with boundary structure. The size definition of the thickness of boundary structure is consistent with its value presented in the experimental results.

Fig. R12. Photograph of (a) the boundary structure and (b) the PM-based ALENS prototype.

Comment: 12) The used excitation voltages in the simulation (400V/1mm or 100 V DC) and that in the experiment (400Vpp shown in Fig. 5) are not consistent. They should be modified to be uniform.

Reply: Thanks for the comment. In the revised manuscript, we uniformed the excitation electric field

to keep same units. We changed “100 V DC or 400V_{pp}” to “400 V/mm or 1600 V/mm” in the revised manuscript.

Comment: 13) The dynamic characteristics of the PM are not provided and discussed in this article. As for an actuation device with multiple DOFs, the dynamic characteristics of each DOF are of concern to determine the practical feasible working condition. For example, the resonant frequency of PM in terms of diverse motion DOFs. The frequency-amplitude characteristics are recommended to test and add into the experiment.

Reply: Thanks for the valuable comment. According to the reviewer’s comment, we tested the output displacement of the PM-based ALENS under different frequencies, and added these data into the experiment part in the revised manuscript. As shown in Fig. R3b, when the working frequency is below 2 kHz, the output displacement of the PM-based ALENS for the AF remains basically stable.

According to this comment, we added the following description in the revised manuscript:

“As shown in **Supplementary Fig. 9**, the impedance spectra have two obvious resonance peaks at 2.9 kHz and 13.8 kHz. When the working frequency is below 2 kHz, the output displacement of the PM-based ALENS for the AF remains basically stable.”

Comment: 14) The application of the PM-based ALENS is presented in the experiment to show the capability of focusing and imaging adjustment. But the presented results are focal length adjustments or image adjustments, not real autofocus operation. Autofocus usually means automatic operation processing. This point should be modified to be more rigorous.

Reply: Thanks for the comment. To fully realize the autofocus, the special circuits and pertinent control algorithm are also required for quickly driving and precision positioning. Herein, we mean that the AF and OIS functions can be realized from a perspective of actuator. To be more rigorous, we changed “Autofocus (AF) function” to “function of adjusting focus (AF)” in the revised manuscript.

Comment: 15) The nonlinear response characteristics of the prototype are all not considered in this

work. As motion devices with multiple motion modes working with a frequency range of 1Hz to 400Hz, the nonlinear hysteresis (in full frequency range) and creep (at low frequency) are two main characteristics affecting the service ability of the PM. Furthermore, the response characteristics of lower than 1Hz and over than 400Hz are also worth considering.

Reply: Thanks for the comment. The nonlinear response characteristics of the prototype were not considered in this work. According to the reviewer's comment, we would like to discuss the output displacement of the PM-based ALENS under different frequencies (from 1 Hz to 8 kHz) and the creep characteristic for the PM-based ALENS.

(1) The output displacement of the PM-based ALENS

As shown in Fig. R3b, when the working frequency is below 2 kHz, the output displacement of the PM-based ALENS for the AF remains basically stable.

(2) The creep characteristic of the PM-based ALENS

Fig. R13 shows the creep curves of the PM-based ALENS, and it can be seen that the variation of displacement is only 0.8 % over 500 s. The result showed the creep characteristic of the PM-based ALENS is minimal.

Fig. R13. Displacement of the PM-based ALENS when driven with a sudden voltage change (under 1000 V/mm). (This figure has been added as Supplementary Fig. 11 in the revised manuscript.)

According to this comment, we added the following description in the revised manuscript:

“As shown in **Supplementary Fig. 9**, the impedance spectra have two obvious resonance peaks at

2.9 kHz and 13.8 kHz. When the working frequency is below 2 kHz, the output displacement of the PM-based ALENS for the AF remains basically stable. ”

“The variation of displacement is only 0.8 % over 500 s, and the creep characteristic of the PM-based ALENS is minimal (**Supplementary Fig. 11**).”

Comment: 16) The ultrahigh strain means that the PM holds low stiffness performance. A question is that the continuous dynamic excitation will cause fatigue in the structure, which may lead to structure damage and further affect the lifetime. How to evaluate the lifetime and characteristic evolution of the PM?

Reply: Thanks for the good suggestion. According to the reviewer’s comment, we would like to discuss the endurance characteristics of the PM-based ALENS. The endurance characteristics of the PM-based ALENS was measured at 2 Hz under 1600 V/mm, as shown in Fig. R4. The inset picture shows transient response of the displacement for the PM-based ALENS, where the output displacements exhibit good repeatability and excellent stability under 1600 V/mm over 14400 vibration cycles, representing its long-term robustness, high reliability and good stability.

According to this comment, we added the following description the revised manuscript:

“Moreover, the output displacement of the PM-based ALENS maintains almost unchanged over 14400 vibration cycles in the durability test, representing its long-term robustness, high reliability and good stability. Detailed analysis can be found in **Supplementary Note 2**.”

Comment: 17) As for the six motion modes of the PM, the positions of generating maximum deformation and node displacement are different. This is unlike other piezoelectric actuation devices that usually have a specific motion output end. In addition, the small size and multiple motion modes of the PM limit the possibility of installing additional execution ends, which seriously limits the application of the PM. These issues are not considered in the article.

Reply: Thanks for the comment. The traditional actuators usually have a specific motion output end, which does not serve AF and OIS functions. The distinctive feature of our designed PM is the multidimensional motions on the surface by controlling different regions of the metasurfaces to

produce the desired motion modes. In our present work, we look forward to realizing multiple motion modes through the smallest possible piezo metasurfaces, providing new design ideas for multi-functions, miniaturization, and integration.

Regarding the reviewer's concern, we believe that it can be solved by designing the piezo metasurfaces with even larger piezoelectric crystals. We didn't put too much attention on installing additional execution ends in the PM for our present work, while we look forward to working with a wide range of partners to deliver on this issue.

Comment: 18) In the method part, how to determine the position of generating maximum displacement and position size is not clarified.

Reply: We thank the reviewer for the valuable comment. According to the reviewer's comment, as shown in Fig. R14, we added the motion points (orange triangles) on the surface of the PM, which has been added as Supplementary Fig. 5 in the revised manuscript.

Fig. R14. Schematic diagram of test positions of the desired motion modes. (This figure has been added as Supplementary Fig. 5 in the revised manuscript.)

Comment: 19) In terms of the experimental condition and setting, the displacement measuring accuracy of the used laser displacement measuring device (LV_S01, a single-point laser vibration testing instrument) is not enough to test the small displacement of the PM in the specific position due to limitation of the device principle. The specialized laser displacement sensor with high measuring accuracy and bandwidth is recommended to continuously measure response characteristic of the PM,

which benefits for showing the full characteristics. Furthermore, the measuring interval of the response under different frequency needs to shorten and the measuring range needs to extend.

Reply: Thanks for the comment. The detection distance, bandwidth, and displacement resolution of the laser rangefinder (LV-S01, SUNNY OPTICAL INSTRUMENTS, China) are from 0.35 m to 20 m, from DC to 25 MHz, and 15 pm, respectively. Moreover, we also added the environmental noise test for the laser rangefinder, including frequency domain analysis and time domain analysis, as shown in Fig. R15. During the test, we kept the environmental noise test below 10 nm. Meanwhile, the output displacements of the PM and PM-based lens are in a range of several nanometers to micrometers, and the measure error is less than 2%. Thus, the laser rangefinder can completely satisfy the testing requirements.

According to the reviewer's comment, we added the output displacement of the PM-based ALENS under different frequencies into the experiment part. As shown in Fig. R3b, when the working frequency is below 2 kHz, the output displacement of the PM-based ALENS for the AF remains stable.

Fig. R15. The environmental noise test for the laser rangefinder includes (a) frequency domain analysis and (b) time domain analysis.

Comment: 20) The experimental results shown in Fig. 3 need to cover wider excitation voltage range, i.e., 0 V to 10 V and over 100 V.

Reply: Thanks for the valuable question. According to the reviewer's comment, we tested and added the output displacement of the PM under different voltages (from 0 to 200 V_{pp}) into the experiment, as shown in Fig. R16. The output displacements remarkably correlated linearly with the operating

voltages at 2 Hz in the range of 0 ~ 200 V_{pp}, which are in good accordance with the simulation data. (We have changed “200 V_{pp}” to “800 V/mm” in the revised manuscript.)

Fig. R16. Experimental verification of the PM with the vibration displacement responses. The experimental setup for measuring the output displacements of the PM. (b-g) The simulated and experimental vibration displacements of the PM in (b) 31-mode, (c) 32-mode, (d) 33-mode, (e) α mode, (f) β mode and (g) λ mode (with LTR = 40) under different electric fields (from 0 V/mm to 800 V/mm) at 2 Hz. (Fig. 3 in the revised manuscript has been updated.)

Comment: 21) From the results shown in Fig. 4a, the PM with electrodes seems to be non-transparent. How does the laser beam penetrate it in the application experiments?

Reply: We thank the reviewer for the question. The Fig. 4a is a schematic diagram of the PM-based ALENS other than the actual PM. In this work, the electrode material is transparent and conductive

ITO (indium tin oxide) films, which allows the light to pass.

Comment: 22) As for the experiment result shown Fig. 5d-f, using the unit of “ μm ” to describe the rotary motions is inappropriate. The angular unit is recommended to show the rotary displacement directly.

Reply: Thanks for the comment. We have changed the description of rotation motion and used rotation degree instead of displacement.

According to the comment, we calculated the rotation angles from the output displacement. As shown in Fig. R17, the rotation angles α , β , and γ are achieved by exciting desired motion modes (the rotary motions around the X-, Y- and Z-axis). Theoretically, the rotation angle (α , β , and γ) of PM generating the rotary motions around the X-, Y- and Z-axis can be calculated by the following formulas,

$$\alpha = \tan^{-1} \frac{L_2/2}{\Delta L_{\text{Rotary-X}}} \quad (3)$$

$$\beta = \tan^{-1} \frac{L_2/2}{\Delta L_{\text{Rotary-Y}}} \quad (4)$$

$$\gamma = \tan^{-1} \frac{L_2/\sqrt{2}}{\Delta L_{\text{Rotary-Z}}} \quad (5)$$

where $\Delta L_{\text{Rotary-X}}$, $\Delta L_{\text{Rotary-Y}}$, and $\Delta L_{\text{Rotary-Z}}$ are the output displacement corresponding to different motion modes; L_2 denote the width of the A_{10} or A_{20} units (shown in Fig. R17).

According to the reviewer’s comments, as shown in Fig. R18d-f, we followed this recommendation to use angle of rotation to describe the rotary motions. Accordingly, we updated Fig. 5 as Fig.R18 in the revised manuscript.

Fig. R17. The output performance of desired motion modes with geometric parameters of various PM by FEM simulation.

Fig. R18. The designed PM-based ALENS and corresponding experiments for vibration responses. (a-f) Pictures on the left are the simulated deformations, including top view and sectional view; Pictures in the middle are vibration amplitudes under different frequencies (from 1 Hz to 400 Hz); Pictures on the right are actuation characteristics under different electric fields (from 120 V/mm to 1600 V/mm), including focal lengths, actuation displacements and rotation angles. (a) the linear motion along the Z-axis (AF), (b) the linear motion along the X-

axis, (c) the linear motion along the Y-axis, (d) the rotary motion around the X-axis (pitch), (e) the rotary motion around the Y-axis (yaw) and (f) the rotary motion around the Z-axis (roll). (Fig. 5 in the revised manuscript has been updated.)

Comment: 23) The more experimental results and discussion to further demonstrate the application value (the capability to solving practical problem) of the PM manipulating the laser beam need to add into the experiments.

Reply: Thanks for the comment. In our present work, our main aim is to introduce a prototype of the PM and perform a corresponding functional verification. The reviewer is correct that it is necessary and valuable to use the PM solving practical problem, which is indeed what we plan to do in the near future. We thank the reviewer again for the constructive suggestion.

Comment: 24) The characteristic comparison listed in the Table 2 is insufficient. The more details about the size, weight, motion end, etc. should be supplemented and discussed.

Reply: Thanks for the good suggestion. Following this suggestion, we made a comparison between the PM-based ALENS and other devices in many aspects, such as dimension, weight, applied electric field, working frequency and hysteresis, as shown in Table R1. The device parameters of PM-based ALENS were comparable to those of the latest researches [Li, Z. *et al.* Adv. Mater. 34, 2107236 (2022); Aktakka, E. E. *et al.* Transducers & Eursensors XXVI, 1583-1586 (2013); Xu, H. *et al.* Microsyst. Technol. 12, 883-890 (2006); S. Chen, *et al.* Precis. Eng. 30, 314-324 (2006); D. Mukhopadhyay, *et al.* Actuat. A-Phys. 147, 340-351 (2008)], as shown in Table R1. Meanwhile, it is worth noting that the most important advantage of the PM-based ALENS is that both AF and OIS functions can be simultaneously achieved by utilizing only one PM.

Table R1 Comparison of the PM-based ALENS with other devices.

Actuation	Dimension (mm×mm)	Weight (g)	Working frequency (Hz)	Hysteresis (%)	Electric field (kV/cm)	Linear motions displacement (μm)			Rotary motions angle (°)			References
						X-axis	Y-axis	Z-axis	X-axis	Y-axis	Z-axis	
	13×13	9.4 g	1 ~ 2000	3.9	16	4.88	5.05	69.8	43.41	44.02	17.90	This work
Piezoelectric	15×15	-	1 ~ 65000	-	18	0.16	0.18	0.22	4	3.5	-	Li et al ¹³
	3×3	-	900	-	20	7.5	7.5	22	60	60	30	Aktakka et al ²⁶
	15×15	-	1400	-	-	1.2	-	1.5	-	-	-	Xu et al ²⁷

Electromagnet ic	D=3 mm	-	4000	-	-	3.0	4.4	3.0	21.6	21.6	30	Chen et al ²⁸
Electrostatic	8×8	-	500	-	17	-	-	18	-	-	103.2	Mukhopadhyay et al ²⁹

(Table 2 in the revised manuscript has been updated.)

According to this comment, we updated the following description in the revised manuscript:

“The device parameters of PM-based ALENS (including dimension, weight, applied electric field, working frequency, hysteresis properties, the displacements and the tilt angles) were comparable to those of the latest researches^{13,26-29}, as shown in **Table 2**. Meanwhile, it is worth noting that the most important advantage of the PM-based ALENS is simple structure, which simultaneously achieve AF and OIS functions by using only one PM.”

Comment: 25) As shown in the Supplementary Fig. 2, the PM is installed in the different boundary structure, a question is how to ensure the installing consistency under multiple operations, which may lead to different fixing stiffness and further affect experimental comparison results.

Reply: Thanks for the comment. It is worth noting that Supplementary Fig. 2 (Supplementary Fig. 5 in the revised manuscript) is the schematic diagram of boundary structure, which are simulation results. For experiments, the boundary structures are prepared by 3D printing technology, and we didn't find any issues in the installing consistency under multiple operations.

Response to reviewer #3

Comment: This article proposed piezoelectric metasurface to simultaneously achieve various vibration modes with ultrahigh piezoelectric strains. They fabricated a piezo metasurface with ten pieces of PIMNT single crystals units, which can simultaneously achieve desired vibration modes and ultrahigh piezoelectric strains in a frequency range of 1-400 Hz. Using piezo metasurface, the authors also designed PM adaptive lens (ALENS) that can simultaneously possess autofocus (AF) and optical image stabilization (OIS) functions. The subject is original and interesting, and it is rich in technical details. The paper can be considered for possible publication subject to the following comments.

Reply: We greatly appreciate the reviewer's positive comments and recommendation very much.

Following are some comments and suggestions:

Comment: 1) Page 4, Line 2, ref 13 should be named and correctly cited.

Page 5, line 7-11, “With applying the AC voltages, the specific region of the PM can generate transverse-extension deformation or axial-bending deformation. Based on the synergistic strain effect among the piezoelectric units, the PM can generate greatly enhanced deformation along a specific direction to achieve the desired vibration modes”, did authors considered structural nonlinearity under such deformation?

Reply: We thank the reviewer for the very careful reading. In our present work, the structure is analyzed within the range of elastic response, which is far away from the region of plastic deformation or nonlinear deformation. As is expected, the simulation results show that the relationship between stress and strain is completely linear when the applied electric fields of the PM increased from 0 to 2000 V/mm (Fig. R19).

Fig. R19. The stress and the strain of the PM in 33-mode under different electric fields (from 0 to 2000 V/mm).

Comment: 2) Page 6, para 2 and 3, and page 7, para 2, have identical sentences and seems more a repetition.

Reply: Thanks for the comments. According to the reviewer's comments, we have revised this section in the revised manuscript to avoid repetition and unclear statement.

Comment: 3) Page 8, last two lines, "It should be noted that if the Young's modulus the boundary is too low it cannot bear the whole weight of the PM". This sentence is not clear and needs to be revised.

Reply: Thanks for the valuable comments. According to this comment, we revised this sentence as shown in below:

"It is worth noting that if the Young's modulus of the boundary structure is too low, it is prone to deformation and thus cannot support the crystals to produce desired motion modes."

Comment: 4) Page 10, line 9-11, "The output displacements remarkably correlated linearly with the operating voltages at 2 Hz in the range of 10 ~ 100 Vpp, which are in good accordance with the simulation data", authors are suggested to explain this sentence. How the output displacement corelated linearly with applied voltage?

Reply: We thank the reviewer for the comment. According to the reviewer's comment, we added the linear fitting curve (black lines) of the output displacement under the different electric fields from 120 V/mm to 1600 V/mm for the PM, as shown in Fig. R20. The results show that the relation between the output displacement and the applied voltage is almost linear.

Fig. R20. The fitting curve of the output displacement under the different electric fields from 120 V/mm to 1600 V/mm.

Comment: 5) The practical significance of this study is unclear. Authors just mentioned that “This research may inspire miniaturization and integration for multi-degrees-of-freedom systems”. This is not enough to catch the interest of industry or applied engineering research community.

Reply: Thanks for the valuable comments. According to the reviewer’s comments, we revised this demonstration in the manuscript:

“This research may inspire miniaturization and integration for multi-degrees-of-freedom systems, such as the micromanipulator for accurate positioning, nano-/micro-electromechanical systems in semiconductor manufacturing and other precision engineering, automatic zoom lens for mobile devices and so on.”

REVIEWER COMMENTS

Reviewer #1 (Remarks to the Author):

The authors have done a very good job in addressing all the comments. I have now been convinced by the novelty and significance of the presented work. However, the experimental demonstration of the PM-based ALENS is still lacking as I could only see the simulations. To show the possibility of using the designed transparent piezoelectric metasurfaces for adaptive optics, some experiments should be added, especially for the PM-based ALENS.

Reviewer #2 (Remarks to the Author):

Thanks to the authors for their detailed responses to my comments and careful modifications to the manuscript. I still have the following concerns on the revised manuscript to discuss with the authors. Surely, this paper can be accepted for publication after some minor revisions.

1. As mentioned in my previous comment 5), the coupling effects among the six motion DOFs are not presented quantitatively. It is suggested to evaluate the displacement responses of other motion DOFs when one of motion DOFs is stimulated.
2. In this round of revision, the author added the new PMs with 9x2 and 21x2 configurations. It is suggested to provide a brief illustration on how to extend the design of PM in the main text.
3. As mentioned in my previous comment 7), the influence evaluation of the main PM structure (especially the sizes) on its motion characteristics are recommended with simulations.
4. According to the responses, please carefully check the supplier of the mentioned universal material testing machine (AI-GS). Japan or Germany?
5. The authors have added a good result for hysteresis properties. In fact, the hysteresis nonlinearity of the piezoelectric device is usually rate-dependent. Therefore, the quantitative evaluation of response hysteresis levels at different frequencies is strongly recommended.

Reviewer #3 (Remarks to the Author):

I may see that the revised paper has addressed all comments in the previous review. This revised article is recommended to the journal for publication.

Responses to Reviewers' comments

[Reviewers' comments are in black; Author responses are in blue; Revisions in the manuscript are highlighted.]

Response to reviewer #1

Comment: The authors have done a very good job in addressing all the comments. I have now been convinced by the novelty and significance of the presented work. However, the experimental demonstration of the PM-based ALENS is still lacking as I could only see the simulations. To show the possibility of using the designed transparent piezoelectric metasurfaces for adaptive optics, some experiments should be added, especially for the PM-based ALENS.

Reply: We appreciate the reviewer's positive comments and recommendation very much. To validate the feasibility of the PM-based ALENS, we performed functional verification test for the PM-based ALENS.

As shown in Fig. R1, we obtained the ray tracing, spot diagram and spot motion variation of the PM-based ALENS via the optical system (Fig. R2).

For the AF function, the parallel light will focus via the PM-based ALENS at activated state. According to the output displacement and the thin-lens equation, it can be obtained that the focal lengths of AF mode declined from ∞ to 35.82 cm as the applied electric field increases from 0 to 1600 V/mm. As shown in Fig. R1a, simulation results show that the parallel beams will converge to the focal point, and the root mean square (RMS) radius (r_{RMS}) in spot diagram decreased from 1520 μm to 137 μm . Similarly, experimental results show that the spot diameter reduce from 780 μm at the non-focusing state to 240 μm at focusing states as the laser passes through the PM-based ALENS.

As the PM-based ALENS generates the pitch, yaw and roll mode, light direction will be rotated around X-, Y- and Z-axis. For example, in the pitch mode, simulation results show that the parallel beams will be rotated around X-axis through the PM-based ALENS, and the spots were obviously deflected to the left or right along X-axis in the diagram (Fig. R1b). In addition, the laser spot will be periodically rotated around X-axis, i.e., the center coordinates of the spot underwent reciprocating motions from (0, -0.205) to (0, 0.203), as illustrated in Fig. R1b.

All of the motions' processes (including AF, pitch, yaw, roll, the coupled modes of the AF and linear motion along X-axis or Y-axis or pitch or yaw or roll) are shown in **Supplementary Movie 1**. The above results show that the PM-based ALENS can be utilized to achieve the AF and OIS functions of optical systems.

Fig. R1. The PM-based ALENS based on the AF and OIS functions under the basic modes and coupled modes for spot motion variation. (a-e) Ray optics simulations (including ray tracing and spot diagram) and dynamic characteristics experiment of optical spot for (a) AF, (b) pitch, (c) roll, (d) in the coupled modes of the AF and pitch and (e) in the coupled modes of the AF and roll in the PM-based ALENS.

Fig. R2. The experimental setup for obtaining the spot size variations of the PM-based ALENS (The optical system is composed of laser, two neutral density filters, a pinhole, a PM-based ALENS and a CCD).

Response to reviewer #2

Comment: Thanks to the authors for their detailed responses to my comments and careful modifications to the manuscript. I still have the following concerns on the revised manuscript to discuss with the authors. Surely, this paper can be accepted for publication after some minor revisions.

Reply: We thank the reviewer for his or her positive recommendation, we have further revised our manuscript and added more results to elaborate our work.

Comment: 1) As mentioned in my previous comment 5), the coupling effects among the six motion DOFs are not presented quantitatively. It is suggested to evaluate the displacement responses of other motion DOFs when one of motion DOFs is stimulated.

Reply: We thank the reviewer for the very valuable comment. We fully agree with the reviewer that it is worthy to further investigate the influence of the coupling effects among various motion modes. In our previous response letter, we mean that our designed multiple modes can be excited at the same time due to non-resonant operation, and thus there is no strong coupling effect like resonant state modes. However, we agree with the reviewer that when several modes are working at the same time, there is a certain coupling phenomenon in the output displacement. We take the displacement responses of linear X-axis motion mode and AF motion mode as an example to analyze the coupling effect on the output displacement (Fig. R3). Here, we assume that the output displacements of linear X-axis motion mode can be stimulated with applied electric fields E_0 while AF motion mode can be simulated with applied electric fields E_1 .

(i) The displacement influence of AF motion mode on linear X-axis motion mode

With an increase in the electric fields E_1 (from 200 V/mm to 1600 V/mm) and E_0 remaining stable ($E_0 = 1600$ V/mm), the displacement declined along X-axis in the coupled mode (Fig. R3d) compare with only linear X-axis motion mode. Within the test range, the maximum output displacement along X-axis in the coupled mode reduced from 4.31 μm to 2.66 μm due to the displacement caused in the opposite direction by the AF motion mode.

(ii) The displacement influence of linear X-axis motion mode on AF motion mode

As the electric fields E_0 increased (from 200 V/mm to 1600 V/mm) while E_1 remained stable ($E_1 = 200$ V/mm), the displacement exhibited a slight decrease along Z-axis in the coupled mode compared

to AF motion mode (Fig. R3e). The maximum output displacement along Z-axis in the coupled mode decreased from 4.29 μm to 4.01 μm due to the displacement generated in the opposite direction by linear X-axis motion mode.

These results suggest that the added motion mode would produce displacements along the opposite direction in the coupled modes, thus reducing the output displacements of the original motion mode. However, the coupling effect of several motion modes can be relieved by more ordered structures with center outward expansion method and circuit control. According to this comment, we will put more attention on the design of the overall structure of the PM in our future study.

Fig. R3. Design of a PM to generate the desired motion modes by FEM simulation. (a-c) The applied voltages and simulated deformations of (a) linear X-axis mode, (b) AF mode, (c) the coupled mode of AF mode and linear X-axis mode. (d) The output displacement along X-axis of linear X-axis mode with AF mode gradually applied. (e) The output displacement along Z-axis of AF mode with linear X-axis mode gradually applied.

Comment: 2) In this round of revision, the author added the new PMs with 9x2 and 21x2 configurations. It is suggested to provide a brief illustration on how to extend the design of PM in the main text.

Reply: Thanks for the very constructive comment. Our proposed PM is arranged in X-Y plane, e.g., 5×2 or 9×2 or 21×2 arrayed structure, where the units are ordered with one in the center while others are arranged with center outward expansion method. As shown in Fig. R4, the center outward expansion method is an orderly structure construction method in which piezoelectric units expand outward circle by circle from the center of the structure with a consideration of required motion modes. Specifically, A_{10} (A_{20}) unit is in the center, A_{11} , A_{12} , A_{13} , and A_{14} (A_{21} , A_{22} , A_{23} and A_{24}) units are the first circle, and A_{15} , A_{16} , A_{17} and A_{18} (A_{25} , A_{26} , A_{27} and A_{28}) units are the second circle, and it is continuously expanded outwards along this way (Fig. R4).

Fig. R4. Schematic diagram of the center outward expansion method. (Supplementary Fig. 1 in the revised manuscript has been updated.)

According to this comment, we updated the following description in the revised manuscript:

“The center outward expansion method is an orderly structure construction method in which piezoelectric units expand outward circle by circle from the center of the structure with a consideration of required motion modes. Specifically, A_{10} (A_{20}) unit is in the center, A_{11} , A_{12} , A_{13} , and A_{14} (A_{21} , A_{22} , A_{23} and A_{24}) units are the first circle, and A_{15} , A_{16} , A_{17} and A_{18} (A_{25} , A_{26} , A_{27} and A_{28}) units are the second circle, and so on (Supplementary Fig. 1)”

“More coupled motion modes can be realized by changing the arrangement, size, or number of piezoelectric units. For example, according to the center outward expansion method, the structure of 5×2 , 9×2 and 21×2 arrayed PMs are shown in Supplementary Fig. 2-4.”

Comment: 3) As mentioned in my previous comment 7), the influence evaluation of the main PM structure (especially the sizes) on its motion characteristics are recommended with simulations.

Reply: Thanks for the valuable comment. The output motion characteristics (including the displacement D_1 and D_3 , the rotation angle α and γ) for the PM as a function of the structure (length to thickness ratio (LTR)) of the PM under the electric field of 400 V/mm was given in Fig. R5a-b. The results show that the effective values (D_1 , α , and γ) increases linearly as the LTR increases, while D_3 increases nonlinearly. The effective values can be controlled over a broad range by resizing the LTR of the PM. Therefore, the researchers can choose the appropriate LTR of the PM according to the output performance requirements.

Fig. R5. Output performance for the desired motion modes as a function of the geometric dimensions of the PM by FEM simulation. (a, b) The displacement (D_j , $j = 1$ and 3) variation of (a) 31-mode and (b) 33-mode with the geometric dimensions of the piezoelectric units for the PM. (c, d) The rotation angle (α and γ) variation of (c) α -mode and (d) γ -mode with the geometric dimensions of the piezoelectric units for the PM. The insets show the geometrical diagrams and deformations of the simulated motion modes by using PM_2 . The black arrows represent displacement directions. The electric field of 400 V/mm was applied on the soft PZT ceramics and [001]-poled PIMNT single crystals.

Comment: 4) According to the responses, please carefully check the supplier of the mentioned universal material testing machine (AI-GS). Japan or Germany?

Reply: Thanks for your careful review. The supplier of universal material testing machine (AI-GS) is from Shimadzu, Japan. We have added this information in the revised manuscript.

Comment: 5) The authors have added a good result for hysteresis properties. In fact, the hysteresis nonlinearity of the piezoelectric device is usually rate-dependent. Therefore, the quantitative

evaluation of response hysteresis levels at different frequencies is strongly recommended.

Reply: We thank the reviewer for the comment. According to the reviewer's comment, we added the hysteresis properties of the PM-based ALENS under the different frequencies (including 2 Hz, 10 Hz and 50 Hz). As shown in Fig. R6, the hysteresis in the ALENS was measured. The calculated results show that the maximum hysteresis at 2 Hz, 10 Hz and 50 Hz are low as 3.9%, 3.8% and 4.2%, exhibiting high control accuracy and high operation reliability.

Fig. R6. The hysteresis characteristic of the PM-based ALENS under the different frequencies, including (a) 2 Hz, (b) 10 Hz, (c) 50 Hz. (This figure has been added as Supplementary Fig. 9 in the revised manuscript.)

According to this comment, we added the following description in the revised manuscript:

“In addition, the hysteresis in the ALENS is the main factor that affects its displacement accuracy, which may affect the properties of the PM-based ALENS. **Supplementary Fig. 8 and 9** depicts the hysteresis of the piezoelectric materials and the PM-based ALENS. The maximum hysteresis of the PM-based ALENS at 2 Hz, 10 Hz and 50 Hz are relatively low, being 3.9%, 3.8% and 4.2% respectively, which indicates that the PM-based ALENS exhibits high control accuracy and operation reliability.”

Response to reviewer #3

Comment: I may see that the revised paper has addressed all comments in the previous review. This revised article is recommended to the journal for publication.

Reply: We thank the reviewer for his/her positive recommendation and his/her efforts on reviewing the manuscript.

REVIEWERS' COMMENTS

Reviewer #1 (Remarks to the Author):

All my comments have been properly addressed. I am happy to accept the manuscript as is.

Reviewer #2 (Remarks to the Author):

The authors have answered my concerns well and revised this paper very carefully, it can be accepted for publication.

Response Letter to Reviewers' Comments

Response to reviewer #1

Comment: All my comments have been properly addressed. I am happy to accept the manuscript as is.

Reply: We thank the reviewer for his or her positive recommendation and his or her efforts on reviewing the manuscript.

Response to reviewer #2

Comment: The authors have answered my concerns well and revised this paper very carefully, it can be accepted for publication.

Reply: We thank the reviewer for his or her positive recommendation and his or her efforts on reviewing the manuscript.